# Learning-Augmented Online Bidding in Stochastic Settings

**Spyros Angelopoulos**

CNRS and International Laboratory

on Learning Systems

Montreal, Canada

spyros.angelopoulos@lip6.fr

**Bertrand Simon**

Université Grenoble Alpes

CNRS, INRIA, Grenoble INP, LIG

Grenoble, France

bertrand.simon@cnrs.fr

## Abstract

Online bidding is a classic optimization problem, with several applications in online decision-making, the design of interruptible systems, and the analysis of approximation algorithms. In this work, we study online bidding under learning-augmented settings that incorporate stochasticity, in either the prediction oracle or the algorithm itself. In the first part, we study bidding under distributional predictions, and find Pareto-optimal algorithms that offer the best-possible tradeoff between the consistency and the robustness of the algorithm. In the second part, we study the power and limitations of randomized bidding algorithms, by presenting upper and lower bounds on the consistency/robustness tradeoffs. Previous works focused predominantly on oracles that do not leverage stochastic information on the quality of the prediction, and deterministic algorithms.

## 1 Introduction

Recent advancements in machine learning have led to the development of powerful tools for efficiently learning patterns in various types of data. These advances spearheaded a novel computational framework, in which the algorithm designer has the ability to integrate a *prediction* oracle into the algorithm's design, theoretical analysis, and empirical evaluation. This paradigm shift has led to the emergence of the field of *learning-augmented* algorithms, which aims to leverage ML approaches towards the development of more efficient algorithms with data-driven capabilities.

Learning-augmented algorithms have witnessed significant growth in recent years, starting with the seminal works [45] and [46]. They have been particularly impactful in the context of sequential and online decision making, in which the algorithm must act on incoming pieces of the input, or adapt its strategy judiciously at appropriately chosen time steps. The defining characteristic of such settings is that the algorithm operates in a state of incomplete information about the input, thus making the application of ML techniques particularly appealing. Specifically, they allow for a more nuanced, and beyond the worst-case performance evaluation than the standard, and often overly pessimistic approaches such as *competitive analysis* [23].

In this work, we focus on a simple, yet important problem of incomplete information, known as *online bidding*. In this problem, faced with some unknown *target* (or "threshold") $u$, a player submits a sequence of bids until one is greater than or equal to $u$, paying the sum of its bids up to that point. Formally, the objective is to find an increasing sequence $X = (x_i)_{i \geq 0}$ of positive numbers of

minimum *competitive ratio*, defined as

$$r(X) = \sup_{u \geq 1} \frac{\text{cost}(X, u)}{u}, \quad \text{where } \text{cost}(X, u) = \sum_{j=0}^{i} x_j : x_{i-1} < u \leq x_i. \tag{1}$$

Despite its seeming simplicity, online bidding has many applications in optimization problems such as incremental clustering and online $k$-median [25], load balancing [13], searching for a hidden target in the infinite line [18], and the design of algorithms with interruptible capabilities [47]; we refer to the survey [28] for a detailed discussion of several applications. As a concrete application, consider the setting in which a user must submit a job for processing in a cloud service: here, the bidding strategy defines the increasing walltimes with which the user submits the job [12]. More broadly, the problem abstracts the process of finding increasingly better estimates of optimal solutions to both online and NP-hard optimization problems, a process that is informally known as "guess and double". It is thus not surprising that online bidding and related competitive *sequencing* problems were among the earliest studied with a learning-augmented lens. We discuss several related results in Section 1.2.

Previous studies of online bidding focused on deterministic settings, i.e., the bidding sequence is output by a deterministic process and, likewise, on deterministic prediction oracles, i.e., oracles that do not provide any probabilistic information on the quality of the prediction. In contrast, very little is known about bidding under *stochastic* settings. There are two main motivating reasons:

• In many realistic applications, the prediction has an inherently distributional nature [30]. For instance, in the related problem of *contract scheduling* [7], the system designer may rely on available historical data about the estimated interruption time (i.e., the target, in the bidding terminology). Hence the question: *Can we find strategies that efficiently leverage a stochastic prediction oracle, while remaining robust against adversarial predictions?*

• It is known that randomization can help improve the competitive ratio of standard online bidding [27]. Randomized algorithms also have additional benefits: they remain much more robust to small prediction errors, and tend to be much less *brittle* than deterministic algorithms, as demonstrated in [33, 21]. Hence the question: *What is the power and the limitations of randomized, learning-augmented bidding algorithms?*

## 1.1 Contribution

Motivated by the discussion above, in this work we study online bidding under two distinct stochastic settings. First, in Section 2 we consider stochastic prediction oracles that provide distributional information on the target. We show how to compute a strategy that attains the best-possible tradeoff between the *consistency* (informally, the expected performance assuming that the target is generated according to the predicted distribution) and the *robustness* (i.e., the performance assuming an adversarial prediction). Using the terminology of learning-augmented computation, we obtain a provably *Pareto-optimal* strategy. Previous work only addressed the case in which the robustness is equal to the best-possible competitive ratio, namely $r = 4$ [7], a case for which optimal strategies have a very simple structure. In our work, instead, we compute the entire Pareto frontier between the consistency and the robustness, which is a much more challenging task since optimal strategies have a complex structure and no clear characterization. Indeed, unlike the deterministic setting where simple geometric strategies suffice, in the stochastic setting such strategies can be extremely inefficient even on trivial distributions. Hence one needs to search a much larger space of candidate strategies which have no obvious structure. To get around this difficulty, we find a suitable "partial" strategy, i.e., one whose bids are within the support of the prediction, minimizes the expected cost, and can be extended to a fully robust, infinite strategy. This can be formulated as a family of linear programs (LPs), whose solution yields the optimal strategy. We also provide a computationally efficient approximation of the Pareto front, with provable performance and run-time guarantees.

Our second main contribution, discussed in Section 3, is the study of randomized bidding strategies. Here, in order to allow a comparison with the known results, all of which have focused on deterministic learning-augmented strategies, we assume a single-valued oracle that predicts the target's value. We first present and analyze a randomized strategy whose consistency/robustness tradeoff provably improves upon the best deterministic one. However, this improvement rapidly dissipates as a function of the robustness requirement $r$. We complement this result with a lower bound that applies to all

randomized algorithms, and which establishes that as $r$ increases, the gap between the deterministic and the randomized consistency rapidly diminishes. Randomization poses new challenges, particularly in the context of negative results: it is not obvious how to show lower bounds on randomized tradeoffs, unlike traditional competitive analysis where Yao's principle [50] is readily applicable, and unlike the deterministic setting in which $r$-robust algorithms have a very clear characterization. Our approach relies on the use of *scalarization*, in that we prove a lower bound on a weighted combination of the two objectives. This is turn allows us to obtain a negative result on their tradeoff, via the minimax theorem (or Yao's principle) [50].

In Section 4, we extend our approaches to other settings and problems. First, we show how our LP-based approach can be applied to a *dynamic* setting in which the oracle provides updated predictions in an on-line fashion, and the (robust) learning-augmented algorithm must maintain optimal consistency relative to each incoming prediction. To our knowledge, this is the first study of competitive sequencing problems under dynamic predictions. Moreover, we show that our results can be extended to the problem of searching for hidden target in the infinite line, which is a well-known problem from search theory and operations research (see, e.g., Chapter 8 in [3]), and extend previous works that did not address robustness issues. We conclude, in Section 5, with an experimental evaluation of our Pareto-optimal algorithm which demonstrates the performance improvements in practice.

## 1.2 Related work

**Learning-augmented algorithms.** Algorithms with predictions have been studied in a large variety of online problems, such as rent-and-buy problems [36], scheduling [41], caching [45], matching [11], packing [37], covering [16] and optimal stopping [32]. This paradigm has also applications outside online computation, e.g., in improving the time complexity of sorting [15], graph problems [14], and data structures [43]. The above are only some representative works and we refer to the online repository [1] for a comprehensive listing. We note that the vast majority of these works do not assume stochastic oracles. The distributional prediction framework was introduced in [30], and has been further applied in the context of search trees [31], online matching [24], and 1-max search [20].

**Pareto-optimal algorithms.** Several studies have focused on the consistency-robustness trade-offs of deterministic learning-augmented algorithms e.g [48, 42, 5, 16, 26, 2]. The study of randomized Pareto-optimality has been limited to problems such as ski rental [49] and search games [10]. For some problems, including online bidding, deterministic Pareto-optimal solutions suffer from brittleness, in that their performance shows marked degradation for very small prediction error [33]. A randomized smoothening can help mitigate brittleness, as shown in [21], but their results still compare to deterministic algorithms. In contrast, in our randomized setting, we compare against randomized algorithms instead of deterministic ones.

**Online bidding and competitive sequencing problems.** For deterministic strategies, a folklore result that goes back to studies of linear search [18] shows that the optimal competitive ratio is equal to 4, and is achieved by a simple strategy of the form $X = (2^i)_{i=0}^{\infty}$. Randomization can help improve the competitive ratio to $e$, whose tightness was proven in [27] using a complex application of dual-fitting to the LP formulation. In regards to learning-augmented algorithms, and single-valued deterministic oracles, [8] gave Pareto-optimal algorithms, whereas [4, 38] gave performance guarantees as a function of the prediction error. Concerning distributional oracles, the only related previous work is [7], which obtained the best consistency of 4-robust strategies. Note that in these studies, Pareto-optimality is attained by simple, geometric strategies of the form $X = (\lambda b^i)_{i \geq 0}$, where $b$ is chosen appropriately as a function of the required robustness $r$. Related competitive sequencing problems such as searching on an infinite line were studied in [5]. No results are known about randomization in learning-augmented bidding, even for single-valued oracles.

## 2 Pareto-optimal bidding with distributional predictions

In this section, we present and analyze deterministic Pareto-optimal algorithms in the distributional prediction model. We begin with the setting in which the prediction $\mu$ is a discrete distribution supported on $k$ points, where point $i$ is defined as $(p_i, \mu_i)$, with $\mu_i \in \mathbb{R}_{\geq 1}$, such that $\sum_{i=1}^{k} p_i = 1$. We assume, without loss of generality, that $\mu_i \leq \mu_j$, for all $i < j$. At the end of the section, we will show how to extend the result to general distributions; we will also discuss time-complexity issues.

The main challenge we face is that, unlike the deterministic setting where simple geometric strategies are optimal, in the stochastic setting such strategies can be extremely inefficient even on trivial distributions, as we show in Proposition 18, Appendix D. Hence one needs to search a much larger space of candidate strategies which have no obvious structure. Our approach consists in restricting the search space to *tight r-extendable partial strategies* (Lemma 5), which are specific finite strategies restricted to the domain of $\mu$ that can be extended to full strategies. We will rely on a linear programming (LP) to find the best partial strategy, by defining the concept of a *configurations* (see Definition 1). By showing that the configuration space is sufficiently restricted, we can find the best strategy by solving the series of LPs that correspond to each possible configuration.

Given a distributional prediction $\mu$ and a strategy $X$, we define the *consistency* of $X$ as

$$\text{cons}(X, \mu) = \frac{\mathbb{E}_{z \sim \mu}[\text{cost}(X, z)]}{\mathbb{E}_{z \sim \mu}[z]}, \tag{2}$$

following [30] and [7]. The robustness of $X$ is the same as its worst-case competitive ratio, and is thus given by (1). By observing that the competitive ratio is maximized for targets which are infinitesimally larger than any of the bids, a strategy $X$ is $r$-robust if and only if it satisfies

$$x_{i+1} \leq r \cdot x_i - \sum_{j \leq i} x_j, \text{ for all } i \geq 0. \tag{3}$$

Hence our problem is formulated as follows: Given a robustness requirement $r \in \mathbb{N}^+$, and prediction $\mu$, we seek an $r$-robust strategy $X$ that minimizes $\text{cons}(X, \mu)$. We will assume that $r$ is constant, independent of other parameters.

The following definition introduces the concept of a *configuration*, which is central in the design and analysis of our strategy, in that it allows us to express the requirements of the problem using linear constraints.

**Definition 1.** *Given a strategy $X$ and a prediction $\mu$, we define the* configuration *of $X$ according to $\mu$ as the vector $(j_1, \ldots, j_k) \in \mathbb{N}^k$ such that*

$$x_{j_i} < \mu_i \leq x_{j_i+1}, \text{ for all } i \in [1, k]. \tag{4}$$

*Furthermore, we say that the vector $\mathbf{j} \in \mathbb{N}^\mathbf{k}$ is a* valid *configuration, relative to a robustness requirement $r$ and prediction $\mu$, if there exists an $r$-robust strategy $X$ with configuration $\mathbf{j}$.*

The number of valid configurations is bounded from above according to the following proposition.

**Proposition 2.** *The number of valid configurations of an $r$-robust strategy is $O(\log^k \mu_k)$.*

Our algorithm will compute a partial, i.e., finite strategy, which must then be extended appropriately to a fully $r$-robust, infinite strategy. The following definition formalizes this requirement.

**Definition 3.** *Given an increasing sequence $Y = (y_i)_{i=0}^l$ that defines a* partial *strategy of $l + 1$ bids, and an increasing sequence $Z = (z_i)_{i=0}^\infty$, we call the strategy with sequence $(y_0, \ldots y_l, z_0, z_1, z_2 \ldots)$ the* extension *of $Y$ based on $Z$. We say that $Y$ is $r$-extendable if there exists a sequence $Z$ such that the extension of $Y$ based on $Z$ is a valid, $r$-robust bidding strategy. Last, we say that $Y$ is* tightly *$r$-extendable if it is $r$-extendable by a sequence $Z$ for which*

$$z_i = rz_{i-1} - \sum_{j=0}^{i-1} z_j - \sum_{j=0}^l y_j, \tag{5}$$

*where $z_{-1}$ is defined to be equal to $y_l$.*

We also formalize what it means for a partial strategy to be $r$-robust:

**Definition 4.** *A partial strategy is* (partially) *$r$-robust if it is $r$-robust assuming the target is smaller than its last bid.*

The following lemma allows us to use tightness as a criterion for extendability.

**Lemma 5.** *A partial strategy $Y = (y_i)_{i=0}^l$ is $r$-extendable if and only if it is tightly $r$-extendable.*

The next lemma provides a necessary and sufficient condition under which a partial strategy is $r$-extendable. Its proof relies on finding the conditions for which a linear recurrence solution that formulates the tight $r$-extension yields an increasing, unbounded infinite sequence.

---

**Algorithm 1** Algorithm for computing the Pareto-optimal strategy.

---

**Input**: Robustness requirement $r$, distributional prediction $\mu = \{(\mu_i, p_i)\}_{i=1}^k$.
**Output:** An $r$-robust strategy of optimal consistency.
  1: For each configuration $\mathbf{j}$, find the solution $x_{\mathbf{j}}$ to the LP $P_{\mathbf{j}}$, if it exists.
  2: Let $\mathbf{x}^* = (x_0^*, \ldots, x_{j_k^*+1}^*)$ be the solution of optimal objective value, for configuration $\mathbf{j}^*$, among all solutions found in Step 1.
  3: Return the strategy

$$(x_0^*, \ldots, x_{j_k^*}^*, \bar{z}_0, \bar{z}_1, \ldots)$$

  where $\bar{Z} = (\bar{z}_i)_{i \geq 0}$ is the tight $r$-extension of $(x_0^*, \ldots, x_{j_k^*}^*)$.

---

**Lemma 6.** *A partial strategy $Y = (y_i)_{i=0}^l$ is $r$-extendable if and only if $Y$ is partially $r$-robust and it satisfies the condition*

$$\frac{\sum_{i=0}^l y_i}{y_l} \leq \frac{r + \sqrt{r(r-4)}}{2}.$$

Algorithm 1 computes the Pareto-optimal strategy. It consists of three steps. In steps 1 and 2, it computes a partial strategy of minimum expected cost, which is partially $r$-robust and is guaranteed to be $r$-extendable. In step 3, it applies the tight $r$-extension, which yields an $r$-robust bidding strategy that is Pareto-optimal. Given a configuration $\mathbf{j} = (j_1, \ldots, j_k)$, we define by $P_{\mathbf{j}}$ the following family of LPs.

$$P_{\mathbf{j}} := \min \quad \sum_{i=1}^k p_i \cdot \sum_{q=0}^{j_i+1} x_q \qquad (6)$$

$$\text{subj. to} \quad x_i \leq r x_{i-1} - \sum_{j=0}^{i-1} x_j, i \in [1, j_k + 1] \qquad (7)$$

$$\sum_{i=0}^{j_k+1} x_i \leq \frac{r + \sqrt{r(r-4)}}{2} x_{j_k+1} \qquad (8)$$

$$x_{j_i} \leq \mu_i \leq x_{j_i+1}, i \in [1, k] \qquad (9)$$

$$x_i \leq x_{i+1}, i \in [0, j_k] \qquad (10)$$

$$x_0 \leq r. \qquad (11)$$

In this family of LPs, the objective function (6) is by definition the expected cost of the partial strategy $(x_0, \ldots, x_{j_k+1})$ assuming a target drawn according to $\mu$, which optimizes the consistency. Constraint 7 guarantees that this partial strategy is $r$-robust, whereas (8) guarantees that the strategy is $r$-extendable, as stipulated in Lemma 6. Constraint (9) describes the configuration $\mathbf{j}$, whereas (11) is an initialization condition that is necessary for monotonicity. The following theorem formalizes the intuition behind the LP formulation, and establishes the correctness of our algorithm.

**Theorem 7.** *Algorithm 1 computes a Pareto-optimal $r$-robust strategy.*

In regards to the time complexity, from Proposition 2 it follows that Step 1 of Algorithm 1 requires solving $O(\log^k \mu_k)$ LPs, each of which takes time polynomial in the size of the input. For constant $k$, the algorithm is polynomial-time, though a realisneuripstic implementation can be time-consuming as $k$ becomes large. To mitigate the effect on the complexity, but also in order to handle any general distribution $\mu$, we can apply a quantization rule to $\mu$ with exponentially spaced levels. This allows to reduce the runtime complexity, at the expense of a small degradation on the approximation of the consistency. The following theorem shows that we can approximate the Pareto frontier to an arbitrary degree of precision in time polynomial in the range of the distributional.

**Theorem 8.** *Let $\mu$ be a distributional prediction with support in $[m, M]$. There exists a quantization of $\mu$ for which Algorithm 1 runs in time polynomial in $\max\{M/m, \log m\}$, and yields a $e^{\frac{1}{c}}$-approximation of the optimal consistency, for any constant $c > 1$ independent of $r$.*

This result is useful especially if the prediction has locality, i.e., $M$ is not vastly larger than $m$. This occurs, for instance, in the related application of contract scheduling, as discussed in [9]. In particular, if $M/m$ is bounded by a constant, the algorithm of Theorem 8 is polynomial-time, hence a PTAS.

## 3 Randomized learning-augmented bidding

In this section, we consider a different stochastic setting. Namely, we do not assume any known stochastic properties of the prediction oracle, but we allow the bidding algorithm access to random bits. That it, the elements of the bidding sequence $X$ are random variables, and so is the cost at which $X$ finds the target. Given a prediction $\hat{u}$ on the target, the consistency and the robustness of the randomized strategy $X$ are defined as

$$\text{cons}(X) = \frac{\mathbb{E}[cost(X, \hat{u})]}{\hat{u}} \quad \text{and} \ \text{rob}(X) = \sup_u \frac{\mathbb{E}[cost(X, u)]}{u}. \tag{12}$$

In the standard setting of no predictions, there is a simple randomized strategy of the form $R = (e^{i+s})_i$, with $s \sim U[0, 1)$, of competitive ratio equal to $e$. This is optimal as shown in [27] using a complex proof based on linear programming and dual fitting techniques. The following proposition shows that the algorithm is essentially tight across all targets, a result which will be useful in interpreting the performance of our learning augmented algorithm.

**Proposition 9.** *The randomized algorithm* $R = (e^{i+s})_i$, *with* $s \in U[0, 1)$ *satisfies* $\mathbb{E}[cost(X, u)]/u = e - O(1/u)$, *for all* $u$.

In the remainder of the section, we provide both upper and lower bounds on the performance of randomized learning-augmented strategies.

### 3.1 Upper bound

We propose a parameterized algorithm with parameters $\delta \in [0, 1)$ and $a > 1$ which, informally, regulate the amount of the desired randomness and the geometric step of the strategy, respectively. Let $\lambda \in [1, a)$, and $j \in \mathbb{N}$ be such that $\hat{u} = \lambda a^{j+\delta}$, namely $j = \lfloor \log_a \hat{u} - \delta \rfloor$, and $\lambda = \frac{\hat{u}}{a^{j+\delta}}$. We define by $R_{\delta,a}$ the randomized strategy with bids $x_i = \lambda a^{i+s}$, where $s \in U[\delta, 1)$. The following theorem quantifies the consistency and the robustness of this strategy.

**Theorem 10.** *Strategy* $R_{\delta,a}$ *has consistency at most* $\frac{a(a-a^\delta)}{a^\delta(a-1)(1-\delta)\ln a}$, *and robustness at most* $\frac{a(a-a^\delta)}{(a-1)(1-\delta)\ln a}$.

$R_{\delta,a}$ interpolates between two extreme strategies. Specifically, for $\delta = 0$, it has consistency and robustness that are both equal to $a/\ln a$, which is minimized for $a = e$. In this case, the strategy is equivalent to the competitively optimal one, and from Proposition 9 it does not exhibit any consistency/robustness tradeoff (it optimizes, however, the robustness). In the other extreme, for $\delta \to 1$, simple limit evaluations show that $\text{rob}(R_{1,a}) = a^2/(a-1)$, whereas $\text{cons}(R_{1,a}) = a/(a-1)$. From the study of deterministic bidding strategies [8], this is the best possible consistency/robustness tradeoff that can be attained by a deterministic strategy (and note that $a^2/(a-1) \geq 4$, for all $a$). This is again expected, since as $\delta \to 1$, the strategy does not use any randomness.

Define now the strategy $R^* = R_{\delta^*, a^*}$ in which the parameters $\delta^*, a^*$ are chosen in optimal manner. Namely, they are the solutions to the optimization problem

$$\max a^\delta \ \text{subject to} \ \frac{a(a - a^\delta)}{(a-1)(1-\delta)\ln a} \leq r, \tag{13}$$

then from Theorem 10, it follows that $R^*$ achieves the best-possible consistency among all $r$-robust strategies in the class $\cup_{\delta,a} R_{\delta,a}$. Hence, $R^*$ has at least as good consistency as the deterministic Pareto-optimal strategy. For instance, for $r = 4$, $R^*$ has consistency approximately equal to 1.724, which is attained at $\delta^* \approx 0.9$. However, as $r$ increases, numerical solutions of (13) show that $\delta^*$ quickly approaches 1. For example, for $r = 4.5$, $\delta^* = 0.95$, and for $r = 5$, $\delta^* \approx 0.99$. This implies that any benefit that $R^*$ attains quickly dissipates as $r$ increases, and randomization is not helpful. In the next section we show that this is not a just a deficiency of $R^*$, but that any randomized strategy has consistency that approaches the best deterministic one, as a function of the robustness $r$.

### 3.2 Lower bound

In this section, we prove a lower bound on the consistency of any randomized $r$-robust strategy, with $r \geq e$. We first begin with an alternative proof that $e$ is the best randomized competitive ratio in

the no-prediction setting. A generalization of this proof will be likewise useful in the prediction setting, namely in the proof of Lemma 12. We adapt an approach due to Gal [35], who studied the randomized linear search problem. For any fixed, bur arbitrarily small $\varepsilon > 0$, define $R = \exp\left(\frac{1-\varepsilon}{\varepsilon}\right)$. Let $D_\varepsilon$ be the distribution on the target with density $\varepsilon/t$, for all $1 \leq t < R$, and has a probability atom of mass $\varepsilon$ at $t = R$. Let $X = (x_i)_{i \geq 0}$ denote a *deterministic* bidding strategy, against the target distribution $D_\varepsilon$. Since the domain of $D_\varepsilon$ is bounded, $X$ is of the form $(x_i)_{i=0}^n$, for some finite $n$. We show that for targets drawn from $D_\varepsilon$, the expected performance ratio of $X$ is at least $(1-\varepsilon)e$. From the minimax theorem, we then obtain that the optimal competitive ratio is arbitrarily close to $e$.

**Theorem 11.** *For any $\varepsilon > 0$ and $D_\varepsilon$ defined as above, it holds that $\mathbb{E}_{u \sim D_\varepsilon}\left[\frac{cost(X,u)}{u}\right] \geq (1-\varepsilon)e$.*

We now move to the learning-augmented setting. Consider again the distribution of targets $D_\varepsilon$, and suppose that the prediction of the oracle is $\hat{u} = R$. Let $X = (x_i)_{i=0}^n$ denote a deterministic strategy, for some $n$ that depends on $\varepsilon$, then the consistency of $X$ is

$$\text{cons}(X) = \frac{cost(X, \hat{u})}{\hat{u}} = \frac{1}{R}\sum_{i=0}^n x_i, \tag{14}$$

which does not depend on distributional assumptions, whereas for the robustness, we have the following useful series of inequalities following the proof of Theorem 11 (see Appendix B):

$$\text{rob}(X) = \varepsilon\sum_{i=0}^n \frac{x_i}{x_{i-1}} \geq \varepsilon(n+1)\left(\prod_{i=0}^n \frac{x_i}{x_{i-1}}\right)^{\frac{1}{n+1}} = (1-\varepsilon)\frac{R^{1/(n+1)}}{\ln R^{1/(n+1)}}. \tag{15}$$

The following is the main technical result that places a lower bound on the scalarized objective.

**Lemma 12.** *Given $\varepsilon > 0$, let $T$ denote the quantity $\frac{R^{\frac{1}{n}}}{\ln R^{\frac{1}{n}}}$. For any $\lambda > 0$, it holds that*

$$rob(X) + \lambda \cdot cons(X) \geq (1-\varepsilon)T + \lambda(1 + \frac{1}{T \cdot F(T)}) - \delta(\varepsilon)$$

*where $F(T) = \ln(T(\ln T + \ln\ln T))$, and $\lim_{\varepsilon \to 0} \delta(\varepsilon) = 0$.*

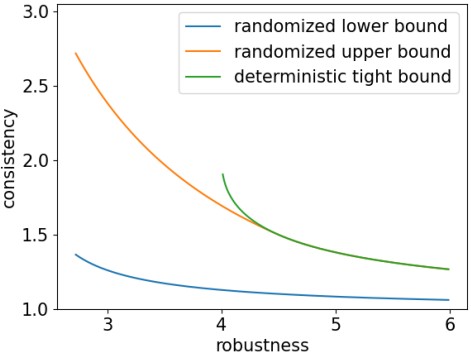

Figure 1: Comparison between the deterministic tight upper bound on the consistency, and the randomized upper and lower bound of Theorems 10 and 13.

We can now prove our lower bound with the help of Lemma 12; specifically, by considering both sufficiently small and sufficiently large values of the parameter $\lambda$.

**Theorem 13.** *Any randomized learning-augmented algorithm of robustness at most $r$ has consistency at least $1 + \frac{1}{(1+\xi)rF(r)}$, for any arbitrarily small, but constant $\xi > 0$.*

Figure 1 compares the consistency of the best deterministic strategy and the lower bound of Theorem 13, as a function of the robustness $r$.

# 4 Further applications

## 4.1 Dynamic predictions

The methodology of Section 2 allows us to obtain Pareto-optimal algorithms for online bidding and related problems, in a dynamic setting in which the oracle can provide multiple predictions in on-line manner. We will use, as illustration, the abstraction of *contract scheduling* [47], a problem that is closely related to online bidding, and which offers a more intuitive interpretation of the dynamic setting. In the standard version of the problem (without predictions), we seek a schedule $X$ that is defined as an increasing sequence $X = (x_i)_{i=0}^{\infty}$. Here, $x_i$ is the length of the $i$-th *contract* that is executed in this schedule, and the objective is to minimize the expression

$$r(X) = \sup_T \frac{T}{\ell(X,T)}, \quad \text{where } \ell(X,T) \text{ is the length of largest contract completed in } X \text{ by time } T.$$

Contract scheduling has several applications in real-time systems and interruptible computation [51, 44, 22]. The learning-augmented version in which a (single) prediction on the interruption $T$ is given ahead of time is identical to bidding, in that both problems have the same Pareto front [9].

Consider now a dynamic version of contract scheduling, where the oracle provides progressively updated, i.e., improved predictions. Let $r$ denote the desired robustness requirement, then our setting can be formulated inductively as follows. Initially, the oracle provides its first prediction on the interruption time, say $\hat{u}_1$, and we seek an $r$-robust deterministic schedule that minimizes the consistency relative to this prediction. Let $X_1$ be this schedule. Inductively, let $X_m$ denote the *updated* schedule that the algorithm decides to follow after the oracle outputs the $m$-th prediction $\hat{u}_m$. Suppose that a new prediction $\hat{u}_{m+1}$, with $\hat{u}_{m+1} > \hat{u}_m$, becomes available at the completion of a contract in $X_m$, say at time $T_m$. The objective is then to define an updated schedule $X_{m+1}$ which starts its execution at time $T_m$, remains $r$-robust, and minimizes the consistency relative to $\hat{u}_{m+1}$.

We can compute each updated schedule $X_m$ by applying an LP-based approach along the lines of Algorithm 1. Moreover, each schedule $X_m$ can be computed efficiently in time $O(\log \hat{u}_m)$. We refer to Appendix C for details and the proof of the following theorem:

**Theorem 14.** *In the dynamic setting, there exists an algorithm for computing an incremental Pareto-optimal schedule of contracts, that runs in time polynomial in the size of each prediction.*

## 4.2 Linear search

The results of Sections 2 and 3 can be extended to another well-known problem, namely searching for a hidden target in the infinite line. This is a fundamental problem from the theory of search games that goes back to works of Beck [17] and Bellman [19], and has been studied extensively in TCS and OR; see [40] for a recent survey. In this problem, a mobile *searcher* initially placed at some point $O$ of the infinite line, must locate an immobile *hider* that hides at some unknown position on the line. The objective is to devise a search strategy $S$ that minimizes the *competitive ratio*, defined by $\sup_h \text{cost}(S,h)/|h|$, where $\text{cost}(S,h)$ is the traveled distance of a searcher that follows $S$ the first time it finds $h$, and $|h|$ is the distance of $h$ from $O$. The best deterministic competitive ratio is equal to 9 [18], whereas the best randomized competitive ratio is $1 + \min_{\alpha>1} \frac{1+\alpha}{\ln \alpha} \approx 4.6$ [35]. Pareto-optimal, deterministic strategies for a single *positional* prediction on the hider were given in [5].

We obtain Pareto-optimal strategies for a distributional prediction on the position of the hider, thus generalizing the result of Alpern and Gal (Section 8.7 in [3]) who gave a strategy of minimum expected search time, but without any robustness guarantees. Furthermore, we can extend the approach of Section 3 to randomized search strategies with a positional prediction. In comparison to online bidding, which is "1-dimensional", the inherent difficulty in linear search is that the searcher operates in a "2-dimensional" space defined by the two halflines. We refer to Appendix C for details.

# 5 Experimental evaluation

We evaluate experimentally our main, Pareto-optimal algorithm, namely Algorithm 1 of Section 2.

**Datasets** The distributional predictions $\mu$ are composed of pairs $(\mu_1, p_1) \dots (\mu_k, p_k)$. We fix $k = 4$, and the support of $\mu$ is in the interval $[1, 10^4]$. The $p_i$ values are either uniform, i.e., all equal to

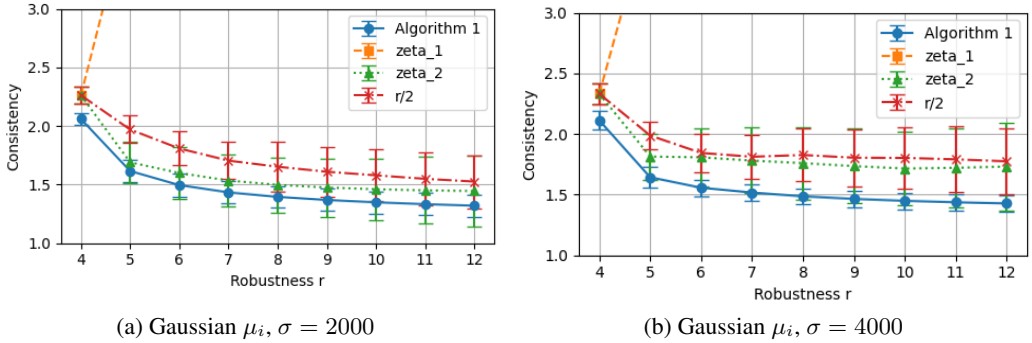

(a) Gaussian $\mu_i$, $\sigma = 2000$           (b) Gaussian $\mu_i$, $\sigma = 4000$

Figure 2: Empirical consistency as a function of the robustness requirement $r$.

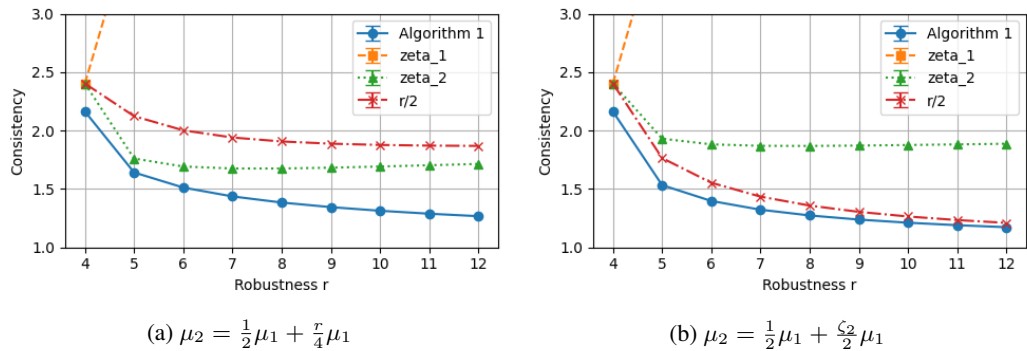

(a) $\mu_2 = \frac{1}{2}\mu_1 + \frac{r}{4}\mu_1$           (b) $\mu_2 = \frac{1}{2}\mu_1 + \frac{\zeta_2}{2}\mu_1$

Figure 3: Empirical consistency when $\mu$ is comprised of two equiprobable points, with $\mu_1 = 5000$ and $\mu_2$ generated as specified in the caption. (a) is adversarial for $\rho = r/2$ and (b) for $\rho = \zeta_2$.

$1/k$, or drawn uniformly in $[0, 1]$ then scaled so that they sum to 1. The values of $\mu_i$ are either drawn uniformly in $[1; 10^4]$; or from a Gaussian distribution centered at the middle of this range, and truncated to the range. This Gaussian distribution is either "narrow", with a standard deviation $\sigma = 2000$, or "spread", with $\sigma = 4000$. In total, we consider two settings for the values of $p_i$ and three setting for the values of $\mu_i$, leading to six different datasets. Furthermore, each dataset is composed of 10 iid samples. We report results on datasets in which the $p_i$ are drawn uniformly and $\mu_i$ follows a Gaussian distribution, and refer to the Appendix D for further experiments and discussion.

**Algorithms** We compare Algorithm 1 to three heuristics of the form $(\lambda \cdot \rho^i)_{i \geq 0}$. Such heuristics are $r$-competitive if and only if $\rho \in [\zeta_1, \zeta_2]$ where $\zeta_1 = \frac{1}{2}(r - \sqrt{r(r-4)})$ and $\zeta_2 = \frac{1}{2}(r + \sqrt{r(r-4)})$ [8]. The three chosen heuristics correspond to $\rho \in \{\zeta_1, \frac{1}{2}r, \zeta_2\}$, that is, to the extrema and the middle of the feasibility range for $\rho$, and $\lambda$ is chosen, for each heuristic, so as to minimize the expected cost over $\mu$. The choice of the base $\rho$ controls how fast the heuristic increases its geometric bids. In particular, the heuristic with $\rho = \zeta_2$ optimizes the consistency assuming a singled-valued prediction [8].

**Results** For each of the 6 datasets, we consider integer values of $r$ within $[4, 12]$ and report, for each algorithm, the average consistency as well as the standard deviation over the 10 samples (error bars). The computation was run on an AMD EPYC 7302 processor using 8 threads. The average runtime for each of the 540 instances was 11.8s. Figure 2 depicts the obtained results.

**Analysis** The experiments show that the heuristic with $\rho = \zeta_1$ is very inefficient, and its consistency increases near-linearly with $r$ (Appendix 5). This is because its bids increase very slowly as a function of $r$. We thus focus on the remaining algorithms. For all $r$, Algorithm 1 has better consistency than all heuristics, thus confirming Theorem 7. It also has better variance, as observed in the error bars, since it outputs optimal solutions to each instance. In contrast, increasing the heuristic base increases their variance, since the bids grow more rapidly, thus rendering the heuristic less flexible. This also explains why the performance gap increases with the standard deviation of $\mu$ in the dataset.

Furthermore, as shown in Figure 3, for each heuristic there exist very simple predictions comprised of only two equiprobable points for which the heuristic has performance that is much inferior to that of our algorithm. This can be explained by the fact that, in the known heuristics, the bids follow a fixed geometric rule and cannot adapt to the specificity of the prediction, even when the latter is extremely simple. In particular, for large $r$, the computations showed that the consistency of Algorithm 1 monotonically approaches 1, whereas the consistency of the heuristics tends to 2. This gap is large, because it implies that the geometric heuristics pay a cost equal to that of for finding the largest point, thus getting no benefit whatsoever from the distributional prediction. In Appendix D we demonstrate that this gap is unbounded for large $r$, even for simple predictions over two points.

## 6 Conclusion

We gave the first systematic study of online bidding in stochastic settings where either the prediction is distributional, or the algorithm is randomized. Our results quantify the power and limitations of randomness, an issue that remains underexplored in the literature, and is bound to become more prevalent in the broader study of learning-augmented algorithms. Our techniques may be applicable to several further extensions, e.g, in searching of a hidden target in a star environment [39] or in layered graphs [34]. They can also apply to settings outside learning-augmented computation, as we demonstrated in Section 4.1. Here, an interesting direction for future work is to apply LP-based approaches towards the study of *barely random* algorithms for online bidding or linear search. For instance, what is the best competitive ratio that can be achieved with only one random bit? This is motivated by recent studies of limited randomness is other well-known problems, e.g., $k$-server [29].

## 7 Acknowledgments

We are thankful to Thomas Lidbetter for helpful discussions on the randomized setting.

This work was supported by the grant ANR-23-CE48-0010 PREDICTIONS from the French National Research Agency (ANR).

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

# Appendix

## A    Proofs from Section 2

*Proof of Proposition 2.* Let $X = (x_i)_{i=0}^{\infty}$ denote an $r$-robust schedule, which thus satisfies (3). From previous studies of this linear recurrence inequality [6], we know that for any $u \geq 1$, there exists an index $l$ such that $x_l \geq u$, where $l = O(\log_{\zeta_1} u)$, and

$$\zeta_1 = \frac{r - \sqrt{r^2 - 4r}}{2}.$$

Note that $\zeta_1$ is constant, since $r$ is assumed to be constant. From Definition 1, it follows that the $i$-th component of a valid configuration has value $O(\log \mu_i)$, which concludes the proof.

We note that this bound is tight, in that from [6] it holds that $l \in \Omega(\log_{\zeta_2}(u))$, where $\zeta_2 = \frac{r + \sqrt{r^2 - 4r}}{2}$.

□

*Proof of Lemma 5.* Suppose that $Y$ is $r$-extendable, then there exists a strategy $Z$ that $r$-extends $Y$. Let $j \in \mathbb{N}$ be the smallest integer such that $z_i < rz_{i-1} - \sum_{j<i} z_j$, and let $\delta = (rz_{i-1} - \sum_{j<i} z_j)/z_i > 1$. Define the strategy

$$Z' = (y_0, \ldots y_l, z_0, \ldots z_{i-1}, \delta z_i, \delta z_{i+1}, \delta z_{i+2}, \ldots)$$

then we remark that $Z'$ is also $r$-robust. Indeed, for $q > i$, we have, as $Z$ $r$-extends $Y$ :

$$\delta z_q \leq \delta(rz_{q-1} - \sum_{j \leq l} y_j - \sum_{j < q} z_j)$$

$$\leq r(\delta z_{q-1}) - \sum_{j \leq l} y_j - \sum_{j=0}^{i-1} z_j - \sum_{j=i}^{q-1} \delta z_j.$$

Hence, $Z'$ is $r$-robust and satisfies (5) for $i = 1$. By repeating this process iteratively, we obtain a sequence that converges, at the limit, to the the tight $r$-extension of $Y$.

For the opposite direction, it suffices to note that a tightly $r$-extendable strategy is $r$-extendable by definition.

□

*Proof of Lemma 6.* Consider a partially $r$-robust partial strategy $Y = (y_i)_{i=0}^{l}$. Note that if $Y$ is not partially $r$-robust, it cannot be $r$-extendable. By scaling all values $y_i$, we can assume, without loss of generality, that $\sum_{i=0}^{l} y_i = r$, as this simplifies the formulas. From Lemma 5, $Y$ is $r$-extendable if and only if it has a tight $r$-extension, say $Z$. We will describe explicitly the constraints that must be imposed on $Z$, in order for the sequence $Y \cup Z$ to be a valid tight $r$-extension of $Y$.

The first bid of $Z$, namely $z_0$, must satisfy the condition $r \cdot y_l = \sum_{i=0}^{l} y_i + z_0 = r + z_0$, hence we have $z_0 = r(y_l - 1)$. The second bid, $z_1$, must satisfy $r \cdot z_0 = \sum_{i=0}^{l} y_i + z_0 + z_1 = r \cdot y_l + z_1$ hence $z_1 = r(z_0 - y_l)$. For each subsequent bid $z_i$, with $i > 1$, it must hold that $z_i = r(z_{i-1} - z_{i-2})$, which can be extended to all $i \in \mathbb{N}$ with the initial conditions $z_{-1} = y_l$ and $z_{-2} = 1$. We have thus defined a linear recurrence relation, whose solution is given by the expression

$$z_{n-2} = \frac{2^{n-1}}{r'} \left( r' \left( (r + r')^n + (r - r')^n \right) - (r - 2y_l) \left( (r + r')^n - (r - r')^n \right) \right), \quad (16)$$

where $r' = \sqrt{r - 4}\sqrt{r} < r$.

The series $z$ describes a valid strategy if and only if it is increasing and unbounded. We consider two cases.

Suppose first that $r' < r - 2y_l$. For $n$ large enough, the terms of the expression of $z_{n-2}$ in $(r + r')^n$ dominate the ones in $(r - r')^n$, so $z_{n-2}$ becomes negative and $z$ is not a valid strategy.

Next, suppose that $r' \geq r - 2y_l$, and we consider two further cases. Suppose first that $r \leq 2y_l$. In this case, both terms of (16) are positive, so $z$ is increasing and a valid strategy. For the next subcase, suppose that $r > 2y_l$, then it follows that the second term of (16) is negative. Since $r' \geq r - 2y_l$, we also have $(r + r')^n + (r - r')^n > (r + r')^n - (r - r')^n$ so $z_{n-2} > \frac{2^{n-1}}{r'}(r' - r + 2y_l)\left((r + r')^n + (r - r')^n\right)$, which is increasing in $n$ and unbounded, so $z$ is a valid strategy.

Consequently, $z$ is a valid strategy if and only if $r' \geq r - 2y_l$, which is equivalent to:

$$2y_l \geq r - r' = r - \sqrt{r(r-4)}$$

$$\Leftrightarrow \frac{\sum_{i=0}^{l} y_i}{y_l} \leq \frac{2r}{r - \sqrt{r(r-4)}} = \frac{2r(r + \sqrt{r(r-4)})}{r^2 - r^2 + 4r} = \frac{r + \sqrt{r(r-4)}}{2}.$$

Therefore, we have shown that a partially $r$-robust strategy $Y = (y_i)_{i=0}^{l}$ admits an $r$-extension $Z$ if and only if $\frac{\sum_{i=0}^{l} y_i}{y_l} \leq \frac{r + \sqrt{r(r-4)}}{2}$, which completes the proof.

In this proof we assumed the general case $r > 4$. For $r = 4$ the recurrence relation is slightly different, since the characteristic polynomial has a double root; we do not show details, since this case is studied in [7]. $\square$

*Proof of Theorem 7.* Let $Y = (y_i)_{i \geq 0}$ denote a Pareto-optimal strategy, i.e., a strategy that is $r$-robust and has minimum consistency. Let $\mathbf{l} = (l_1, \ldots l_m)$ be the configuration of $Y$ according to $\mu$. Since $Y$ is $r$-robust, the sequence $Y' = (y_i)_{i=0}^{l_m}$ is partially $r$-robust, as well as $r$-extendable. From Lemma 6, this implies that $Y'$ satisfies all the LP constraints (7), (8) and (11). Hence, in Step 2, Algorithm 1 computes a partial strategy $x^*$ for some configuration $\mathbf{j}^*$ for which $C := \sum_{i=1}^{k} p_i \sum_{q=0}^{j_i+1} x_q^*$ is at most the consistency of $Y$. Furthermore, since $x^*$ satisfies (8), from Lemmas 5 and 6 it is aggressively $r$-extendable. Therefore, we conclude that the strategy returned by the algorithm is $r$-robust, and has consistency at most that of $Y$, hence it is Pareto-optimal. $\square$

*Proof of Theorem 8.* Let $F_\mu$ denote the cumulative distribution function of the prediction $\mu$. Since $\mu$ has support in the interval $[m, M] \subset \mathbb{R}_{>1}$, we have $F_\mu(m) = 0$ and $F_\mu(M) = 1$. We can assume, for simplicity, that $M > e \cdot m$ hence $M/m = \Omega(1)$, as we focus on the asymptotic complexity of the algorithm.

We define the distribution $\bar{\mu}$ as a quantization of $\mu$ into $k = \lceil c \log(M/m) \rceil$ points (or "levels") $\bar{\mu}_1 \ldots \bar{\mu}_k$ of value $\bar{\mu}_i = m \cdot e^{i/c}$. Note that $\bar{\mu}_k \geq m \cdot e^{\log(M/m)} \geq M$. The probability $p_i$ associated with $\bar{\mu}_i$ is defined to be equal to $F(\bar{\mu}_i) - F(\bar{\mu}_{i-1})$ (assuming $\mu_0 = m$). Intuitively, $\bar{\mu}$ is obtained from $\mu$ by shifting probability mass to the right, by a maximum multiplicative factor of $e^{1/c}$, and results in $O(\log(M/m))$ points.

Consider running Algorithm 1 on $\bar{\mu}$. The space of all configurations is now more restricted, in comparison to the setting of Section 2. Specifically, as discussed in the proof of Proposition 2, we know that for any $r$-robust strategy $X = (x_i)_{i=0}^{\infty}$, the first index $i$ such that $x_i \geq \bar{\mu}_1 = m \cdot e^{1/c}$ must satisfy $i = O(\log m)$. Hence, we can restrict the execution of Algorithm 1 on configurations $\mathbf{j} = (j_1, \ldots, j_k)$ for which $j_1 = O(\log m)$. Similarly, we can also restrict $\mathbf{j}$ for which $j_k - j_1 = O(\log(M/m))$. This results in a total of configurations that is at most

$$O\left(\log m \cdot \prod_{i=2}^{k}(j_i - j_{i-1})\right).$$

We now bound $\prod_{i=2}^{k}(j_i - j_{i-1})$ using the fact that there exists a constant $C \geq c$ for which $\sum_{i=2}^{k}(j_i - j_{i-1}) = j_k - j_1 \leq C\lceil \log(M/m) \rceil$, and the AM-GM inequality:

$$\prod_{i=2}^{k}(j_i - j_{i-1}) \leq \left(\frac{j_k - j_1}{k-1}\right)^{k-1} \qquad \text{(AM-GM inequality)}$$

$$\leq \left(\frac{C\lceil \log \frac{M}{m}\rceil}{\lceil c \log \frac{M}{m}\rceil - 1}\right)^{\lceil c \log \frac{M}{m}\rceil - 1} \qquad \text{(definition of } k \text{ and upper bound on } j_k - j_1)$$

$$\leq \left(\frac{2C\log \frac{M}{m}}{\frac{c}{2}\log \frac{M}{m}}\right)^{2c \log \frac{M}{m}} \qquad \text{(follows from } \log \frac{M}{m} > 1 \text{ and } C > c > 1)$$

$$\leq \left(\frac{4C}{c}\right)^{2c \log \frac{M}{m}} \leq \left(\frac{M}{m}\right)^{2c \log \frac{4C}{c}} = \left(\frac{M}{m}\right)^{O(1)}.$$

The complexity to solve each LP in Algorithm 1 is polynomial in the number of variables and constraints, so polynomial in $k = O(\log(M/m))$ and $j_k = O(\log m + \log(M/m))$, hence the total complexity is polynomial in $O(\frac{M}{m} + \log m)$.

Regarding the consistency of the strategy $X$ returned by this procedure, we have:

$$\text{cons}(X, \mu) = \frac{\mathbb{E}_{z \sim \mu}[\text{cost}(X, z)]}{\mathbb{E}_{z \sim \mu}[z]} \qquad \text{(definition of consistency)}$$

$$\leq \frac{\mathbb{E}_{z \sim \bar{\mu}}[\text{cost}(X, z)]}{\mathbb{E}_{z \sim \mu}[z]} \qquad \text{(follows from the definition of } \bar{\mu} \text{ and monotonicity of cost)}$$

$$\leq \frac{\text{cons}(X, \bar{\mu}) \cdot \mathbb{E}_{z \sim \bar{\mu}}[z]}{\mathbb{E}_{z \sim \mu}[z]} \qquad \text{(definition of consistency)}$$

$$\leq \text{cons}(X, \bar{\mu}) \cdot e^{1/c}. \qquad \text{(follows from the definition of } \bar{\mu})$$

The first inequality comes from the fact that probability masses have only be moved to the right from $\mu$ to $\bar{\mu}$ and the function $z \mapsto \text{cost}(X, z)$ is non-decreasing. The second is the definition of $\text{cons}(X, \bar{\mu})$. And the last one comes from fact that no probability mass has been shifted by a multiplicative factor larger than $e^{1/c}$ from $\mu$ to $\bar{\mu}$.

Therefore, as Algorithm 1 offers the best consistency / robustness tradeoff, this procedure results in consistency that is within a factor at most $e^{1/c}$ from the best consistency achievable in the worst-case, assuming robustness $r$. $\qquad \square$

## B  Proofs from Section 3

*Proof of Proposition 9.* Let $\delta \in [0, 1)$ be such that for the unknown target $u$, it holds that $u = e^{j+\delta}$, where $j \in \mathbb{N}$. Then $X$ will either locate $u$ in iteration $j$ with probability $1 - \delta$, or in iteration $j + 1$ with probability $\delta$. Hence

$$\mathbb{E}[\text{cost}(X, u)] = \frac{1}{e^{j+\delta}}\left(\mathbb{E}[\frac{e^{j+1+s}-1}{e-1}|s \geq \delta]\Pr[s \geq \delta] + \mathbb{E}[\frac{e^{j+2+s}-1}{e-1}|s \leq \delta]\Pr[s \leq \delta]\right)$$

$$= \frac{e}{e^\delta(e-1)}\left(\mathbb{E}[e^s|s \geq \delta](1-\delta) + e\mathbb{E}[e^s|s \leq \delta]\delta\right) - O(\frac{1}{u})$$

$$= \frac{e}{e^\delta(e-1)}\left(\frac{e-e^\delta}{1-\delta}(1-\delta) + \frac{e(e^\delta-1)}{\delta}\delta\right) - O(\frac{1}{u})$$

$$= e - O(\frac{1}{u}),$$

which completes the proof. $\qquad \square$

*Proof of Theorem 10.* To evaluate the consistency, note that $R_{\delta,a}$ locates $\hat{u}$ at iteration $j$, regardless of the random outcome of $s$. Therefore,

$$\mathbb{E}[\text{cost}(R_{\delta,a}\hat{u})] = \lambda \mathbb{E}[\frac{a^{j+1+s}-1}{a-1}] \leq \lambda a^{j+1} \frac{a-a^{\delta}}{(1-\delta)\ln a},$$

which establishes the consistency of $R_{\delta,a}$, since $\hat{u} = \lambda a^{j+\delta}$.

To bound the robustness, let $u$ denote an arbitrary target, and let $\mu \in [0,1)$ be such that $u = \lambda a^{k+\mu}$, where $k \in \mathbb{N}$. We consider two cases:

*Case 1: $\mu \leq \delta$:* In this case, $R_{\delta,a}$ locates $u$ at iteration $k$. Hence, similarly to the calculations above, we obtain that

$$\text{rob}(R_{\delta,a}) = \sup_{\mu \in (0,\delta]} \frac{\mathbb{E}[\text{cost}(R_{\delta,a},u)]}{\lambda a^{k+\mu}} \leq \sup_{\mu \leq \delta} \frac{a^{k+1}(a-a^{\delta})}{a^{\mu+k}(a-1)(1-\delta)\ln a} = \frac{a(a-a^{\delta})}{(a-1)(1-\delta)\ln a}.$$

*Case 2: $\mu > \delta$.* In this case, $R_{\delta,a}$ locates the target either at iteration $k$, if $s \geq \mu$, or at iteration $k+1$, otherwise. Hence

$$\text{rob}(R_{\delta,a}) = \frac{1}{\lambda a^{k+\mu}}\left(\mathbb{E}[\lambda\frac{a^{k+s+1}}{a-1}|s \geq \mu]\Pr(s \geq \mu) + \mathbb{E}[\lambda\frac{a^{k+s+2}}{a-1}|s < \mu]\Pr(s < \mu)\right)$$

$$= \frac{1}{(a-1)(1-\delta)\ln a}(a-a^{\mu}+a(a^{\mu}-a^{\delta}))$$

$$\leq \frac{a(a-a^{\delta})}{(a-1)(1-\delta)\ln a}.$$

$\square$

*Proof of Theorem 11.* Let $G$ denote the cdf of $D_{\varepsilon}$. We also define $x_{-1} = 1$, for initialization purposes. Since any target drawn from $D_{\varepsilon}$ is bounded by $R$, we can assume that $X$ is of the form $X = (x_i)_{i=0}^n$, for some $n$ such that $x_n = R$. We have that

$$\mathbb{E}_{u \sim D_{\epsilon}}[\frac{\text{cost}(X,u)}{u}] = \sum_{i=0}^n \int_{x_{i-1}}^{x_i} \sum_{j=0}^i x_i \frac{1}{y}dG(y) \qquad \text{(by definition)}$$

$$= \sum_{i=0}^n x_i \int_{x_{i-1}}^R \frac{1}{y}dG(y) \qquad \text{(rearranging terms)}$$

$$= \varepsilon \sum_{i=0}^n \frac{x_i}{x_{i-1}} \qquad \left(\int_{x_{i-1}}^R \frac{1}{y}\,dG(y) = \varepsilon\left(\int_{x_{i-1}}^R \frac{1}{y^2}\,dy + \frac{1}{R}\right)\right)$$

$$\geq \varepsilon(n+1)\left(\prod_{i=0}^n \frac{x_i}{x_{i-1}}\right)^{\frac{1}{n+1}} \qquad \text{(AM-GM inequality)}$$

$$= (1-\varepsilon)\frac{R^{\frac{1}{n+1}}}{\ln R^{\frac{1}{n+1}}} \qquad \text{(from the definition of $R$ and telscoping product)}$$

$$\geq (1-\varepsilon)e. \qquad \text{(since $x/\ln x$ is minimized at $x = e$)}$$

$\square$

*Proof of Lemma 12.* Let us denote by $\rho_i$ the ratio $x_i/x_{i-1}$ and by $m$ the ratio $1/n$.

From (15) it follows that

$$\mathbb{E}[\text{rob}(X) + \lambda\text{cons}(X)] = \varepsilon \sum_{i=0}^n \rho_i + \lambda + \frac{\lambda}{\rho_n} = \varepsilon \sum_{i=0}^{n-1} \rho_i + \varepsilon\rho_n + \lambda + \frac{\lambda}{\rho_n}. \qquad (17)$$

For further convenience of notation, let us denote $a := \rho_n$, then from the AM-GM inequality and a reasoning along the lines of (15), Eq. (17) gives

$$\mathbb{E}[\text{rob}(X) + \lambda\text{cons}(X)] \geq (1 - \varepsilon)\frac{x_{n-1}^{\frac{1}{n}}}{\ln R^{\frac{1}{n}}} + \epsilon a + \lambda + \frac{\lambda}{a}$$

$$= (1 - \varepsilon)T\frac{1}{a^{\frac{1}{n}}} + \epsilon a + \lambda + \frac{\lambda}{a} \qquad (18)$$

by recalling that $T$ denotes the quantity $\frac{R^{\frac{1}{n}}}{\ln R^{\frac{1}{n}}}$.

Define $f(a) := (1 - \varepsilon)T\frac{1}{a^{\frac{1}{n}}} + \varepsilon a + \frac{\lambda}{a}$. We will prove that as $\varepsilon \to 0$, $f$ has a minimum that is arbitrarily close to $f(a^*)$, where

$$a^* = \frac{m(1 - \varepsilon)}{\varepsilon}T. \qquad (19)$$

To see why this suffices to complete the proof, note first that

$$R^m \leq e^{W_0(T)},$$

from the definition of the Lambert function, and more precisely its right branch. From the definition of $R$, the above inequality implies that

$$m \leq \frac{\epsilon}{1 - \varepsilon}W_0(T), \qquad (20)$$

and note that

$$\lim_{\varepsilon \to 0} m = 0. \qquad (21)$$

From (19) and (20) it will then follow that

$$a^* \leq T \cdot W_0(T).$$

Moreover, as $\varepsilon \to 0$, $f(a)$ is arbitrarily close to the function

$$g(a) := (1 - \varepsilon)T + \varepsilon a + \frac{\lambda}{a},$$

hence $f(a^*)$ is arbitrarily close to $g(a^*)$, and from (18) this completes the proof. Note that $\delta(\epsilon)$ can be defined as a function that is comprised by infinitesimally small terms, due to the approximations that we will introduce in what follows.

Next, we find the extreme points of $g(a)$. The derivative of $g$ with respect to $a$ is equal to

$$g'(a) = -(1 - \varepsilon)mTa^{-m-1} + \varepsilon - \frac{\lambda}{a^2}.$$

To find the local extreme points, we solve $g'(a) = 0$. Since $g'$ is continuous, as $\varepsilon \to 0$, local extrema are arbitrarily close to the solution of the equation

$$-(1 - \varepsilon)mT\frac{1}{a} + \varepsilon - \frac{\lambda}{a^2} = 0.$$

The solution $a^*$ of this quadratic equation, considering that $a^* > 0$ is

$$a^* = \frac{(1 - \varepsilon)mT + \sqrt{((1 - \varepsilon)mT)^2 + 4\varepsilon\lambda}}{2\varepsilon},$$

and note that as $\varepsilon \to 0$, $a^*$ approaches $(1 - \varepsilon)mT/\varepsilon$, hence establishing (19). Last, this value corresponds to the unique maximum of $g$. This is because $g''(a) = m(1 - \epsilon)T\frac{1}{a^2} + \frac{\lambda}{a^3} > 0$. $\qquad \square$

*Proof of Theorem 13.* By way of contradiction, suppose that there exists a randomized $r$-robust strategy $X$ with consistency at most $C = 1 + 1/(rW_0(r))$, then for every $\lambda > 0$

$$\text{rob}(X) + \lambda\text{cons}(X) \leq r + \lambda C. \qquad (22)$$

Define
$$S = \frac{R^{\frac{1}{n+1}}}{\ln R^{\frac{1}{n+1}}},$$
then recalling the definition of $T$, we have

$$T \le R^{\frac{1}{n(n+1)}} S. \tag{23}$$

Let $\xi > 0$ be any fixed constant. We consider two cases:

*Case 1:* $S \ge (1 + \xi)r$. In this case, from (15) we have that $\mathbb{E}[\mathrm{rob}(X)) \ge (1 + \xi)r$. This, however, contradicts Yao's principle from (22) and any sufficiently small value of $\lambda$.

*Case 2:* $S < (1 + \xi)r$. From (23) and (21) we obtain that $T$ is arbitrarily close, but smaller than $S$, as $\epsilon \to 0$. For sufficiently large $\lambda$, e.g. $\lambda = r^2$, Lemma 12 yields

$$\mathbb{E}[\mathrm{rob}(X)) + \lambda \mathrm{cons}(X)] \ge (1 - \varepsilon)T + r^2 \left(1 + \frac{1}{TW_0((T)}\right)$$
$$\ge (1 - \varepsilon)(1 + \xi)r + r^2 \left(1 + \frac{1}{(1 + \xi)rW_0((1 + \xi)r)}\right),$$

where the second inequality follows from the monotonicity of the RHS, assuming that $T < r^2$ (otherwise, the lower bound follows trivially). $\qquad\square$

## C  Proofs and omitted details from Section 4

### C.1  Contract scheduling with dynamic predictions

We show how to compute, inductively, each updated schedule in the dynamic setting. Recall that the fist schedule, $X_1$, can be computed as the Pareto-optimal schedule for prediction $\hat{u}_1$ [9]. Let $X_m$ denote the schedule computed for prediction $\hat{u}_m$; from the discussion of the setting in Section 4.1, $X_m$ begins the execution of its first contract at time $T_{m-1}$.

We describe how to compute schedule $X_m$, for prediction $\hat{u}_m$. From Proposition 2, any $r$-robust schedule can complete at most $O(\log \hat{u}_m)$ contracts by time $\hat{u}_m$. Therefore, it suffices to solve the following family of LPs $P_j$, for $j \in O(\log \hat{u}_m)$:

$$P_{\mathbf{j}} := \max \quad x_j \tag{24}$$

$$\text{subj. to} \quad x_i \le rx_{i-1} - \left(T_{m-1} + \sum_{k=0}^{i-1} x_k\right), i \in [1, j] \tag{25}$$

$$T_{m-1} + \sum_{i=0}^{j} x_i \le \hat{u}_m \tag{26}$$

$$T_{m-1} + \sum_{i=0}^{j} x_i \le \frac{r + \sqrt{r(r-4)}}{2} x_j \tag{27}$$

$$x_i \le x_{i+1}, i \in [0, j-1] \tag{28}$$

$$x_0 \le r \cdot \mathtt{last}(X_{m-1}) - T_{m-1}. \tag{29}$$

The objective of the LP is to maximize $x_j$: here, $j$ is the index of the largest contract that completes by time $\hat{u}_m$. Maximizing $x_j$ minimizes the consistency of the schedule. Constraint (25) guarantees the $r$-robustness of the schedule: this is because the worst-case interruptions occur right before the completion of a contract [47]. Note that the time starts at $T_{m-1}$, which is the time that $\hat{u}_m$ is available and the first contract of $X_m$ will be scheduled. Constraint (26) states that $x_j$ is indeed completed by time $\hat{u}_m$. Constraint (27) guarantees that the final schedule can be $r$-extendable, and constraint 28 captures the monotonicity of the contracts. Last, (29) is an initialization constraint. Here, the notation

$\texttt{last}(X_{m-1})$ denotes the last contract that was executed in $X_{m-1}$ before $\hat{u}_m$ was issued, i.e., the last contract executed before $T_{m-1}$.

To complete the computation of $X_m$, after solving the above family of LPs and computing a partial schedule of the form $(x_0^*, \ldots x_{j^*}^*)$ (for the optimal index $j^*$), it suffices to augment this partial schedule with its tight $r$-extension, as in step 3 of Algorithm 1.

## C.2 Linear search

We discuss how the approach of Sections 2 and 3 can be applied to the linear search problem.

### C.2.1 Pareto-optimal search

Let us assume the convention that the left half-line (i.e., the half line to the left of $O$) is mapped to the negative reals, whereas the right halfline is mapped to the positive reals. A search strategy $X$ can be described as a sequence of the form

$$X = ((-1)^{i+\delta} \cdot x_i)_{i \geq 0}, \quad \text{where } x_i > 0 \text{ and } \delta \in \{0, 1\}.$$

Consider a target hiding at the point $(-1)^{\delta'} u$ where $\delta' \in \{0, 1\}$. Then $X$ discovers the target at cost

$$\text{cost}(X, u) = u + 2 \sum_{j=0}^{i-1} x_j : x_{i-2} < u \leq x_i \text{ and } (-1)^{\delta+i} = (-1)^{\delta'},$$

where the factor 2 accounts for the searcher moving away from $O$, then returning to $O$, in each iteration.

With the above definitions in place, we can define configurations in a manner similar to Definition 1. For a given prediction $\mu = (\mu_1, p_1), \ldots, (\mu_k, p_k)$ with $\mu_i \in \mathbb{R}$, we first list the orderings in which a search strategy may find each $\mu_i$, which are at most $2^k$. Indeed, initially and after each $\mu_i$ is found by the searcher, there are at most two possibilities for the next $\mu_j$: the one directly to the right of $\mu_i$ or to its left. Let $\sigma$ be such a valid ordering, so $X$ first finds $\mu_{\sigma(1)}$, then $\mu_{\sigma(2)}$ until $\mu_{\sigma(k)}$. A configuration of $X$ according to $\mu$ and $\sigma$ is defined as the vector $(j_1, \ldots, j_k) \in \mathbb{N}^k$ such that, for all $i \in [1, k]$:

$$(-1)^{j_i - 1 + \delta} x_l < \mu_{\sigma(i)} \leq (-1)^{j_i + 1 + \delta} x_{j_i + 1} \quad \text{if } \mu_{\sigma(i)} > 0$$
$$(-1)^{j_i - 1 + \delta} x_l > \mu_{\sigma(i)} \geq (-1)^{j_i + 1 + \delta} x_{j_i + 1} \quad \text{if } \mu_{\sigma(i)} < 0.$$

Furthermore, we note that a strategy is $r$-robust if and only if for all $i$ it holds that

$$x_{i+1} \leq \frac{r-1}{2} \cdot x_i - \sum_{j \leq i} x_j.$$

Note that the definition is matching the one for bidding, by replacing $r$ with $\rho := \frac{r-1}{2}$. Hence, the tight $r$-extension of a strategy has the same properties as in Lemma 5. Therefore, Lemma 6 holds as well, by replacing $r$ with $\rho$. It follows that a partial strategy $Y = ((-1)^{i+\delta} y_i)_{i=0}^{\ell}$ is $r$-robust if and only if it is partially $r$-robust and $\sum_{i=0}^{\ell} \frac{y_i}{y_\ell} \leq \frac{1}{2}\rho + \frac{1}{2}\sqrt{\rho(\rho-4)}$.

We can therefore design an algorithm analogous to Algorithm 1, which first lists all possibilities for $\delta \in \{0, 1\}$, $\sigma$, and the configurations. Then, it finds the best corresponding $r$-robust partial solution which can be $r$-extended, if it exists. This computation is done through a family of LPs similar to that for the bidding problem:

$$P_{\mathbf{j},\sigma,\delta} := \min \quad \sum_{i=1}^{k} p_i \cdot \left( |\mu_{\sigma(i)}| + 2\sum_{q=0}^{j_i} x_q \right)$$

$$\text{subj. to} \quad x_i \leq \rho \cdot x_{i-1} - \sum_{j=0}^{i-1} x_j, i \in [1, j_k + 1]$$

$$\sum_{i=0}^{j_k+1} x_i \leq \frac{\rho + \sqrt{\rho(\rho-4)}}{2} x_{j_k+1}$$

$$(-1)^{\delta+j_i-1} x_{j_i-1} \leq \mu_{\sigma(i)} \leq x_{j_i+1}, i \in [1, k] \text{ and } \mu_{\sigma(i)} > 0$$

$$(-1)^{\delta+j_i-1} x_{j_i-1} \geq \mu_{\sigma(i)} \geq -x_{j_i+1}, i \in [1, k] \text{ and } \mu_{\sigma(i)} < 0$$

$$x_i \leq x_{i+2}, i \in [0, j_k]$$

$$x_i \geq 0, i \in [0, j_{k+1}]$$

$$x_0 \leq r.$$

### C.2.2 Randomized search

We now show how to extend the approach of Section 3 to the problem of linear search.

First, for the standard competitive analysis without predictions, we note that there are two different ways of defining and analyzing randomized strategies of optimal competitive ratio. The first concerns the setting with the assumption that the target cannot hide at a distance less than unit from $O$ (otherwise no constant-competitive strategies exist). Here, efficient strategies are of the form[1] $X = ((-1)^b a^{i+s})_{i=0}^{\infty}$, where $b$ is a random bit in $\{0, 1\}$, and $s$ is chosen u.a.r in $[0, 1]$. The second setting assumes that the search can start with infinitesimally small oscillations around $O$: this "bi-infinite" setting does not require a random starting choice between the left and right halflines, and an efficient strategy if of the form $X = ((-1)^i a^{i+s})_{i=-\infty}^{+\infty}$, where $s$ is chosen u.a.r. in $[0, 2]$. Both settings lead to the same optimal competitive ratio of

$$q := 1 + \min_{a>1} \frac{1+a}{\ln a} \approx 4.6. \tag{30}$$

For convenience, in what follows we will assume the bi-infinite model, under which the cost expressions are easier to derive.

**Upper bound** We first obtain a randomized strategy, parameterized over $\delta \in [0, 2]$ and $a > 1$. Let $\hat{u}$ denote the predicted target, and let $j \in \mathbb{N}, \lambda \in [1, a)$ be such that $|\hat{u}| = \lambda a^{j+\delta}$. Suppose also, without loss of generality that $\hat{u}$ is in the right halfline. We define $R_{\delta,a}$ as the randomized strategy $((-1)^{d+i} \lambda a^{i+s})_i$, where $s$ is chosen u.a.r. in $[\delta, 2)$, and $d$ has the same parity as $j$.

**Theorem 15.** $R_{\delta,a}$ *has consistency at most* $1 + 2\frac{a^2-a^\delta}{a^\delta(2-\delta)(a-1)\ln a}$ *and robustness at most* $1 + 2\frac{a^2-a^\delta}{(a-1)(2-\delta)\ln a}$.

*Proof.* By definition, the strategy discovers $\hat{u}$ at iteration $j$, hence its consistency is bounded as follows:

$$\text{cons}(R_{\delta,a}) = 1 + 2\frac{a^j}{a-1}\frac{1}{a^{j+\delta}}\mathbb{E}[e^s]$$

$$= 1 + 2\frac{1}{(a-1)a^\delta}\frac{a^2-a^\delta}{(2-\delta)\ln a}$$

$$= 1 + 2\frac{a^2-a^\delta}{a^\delta(2-\delta)(a-1)\ln a}.$$

---

[1]This setting is analyzed in [Kao, Ming-Yang, John H. Reif, and Stephen R. Tate. "Searching in an unknown environment: An optimal randomized algorithm for the cow-path problem." Information and computation 131.1 (1996): 63-79]. The work is a rediscovery of an original result due to Gal [35].

To bound the robustness, we can generalize the competitive analysis of the randomized $q^*$-competitive strategy. More precisely, following the lines of the proof of Lemma 8.5 in [3] we obtain that

$$\text{rob}(R_{\delta,a}) = 1 + \frac{a^2}{a-1} \int_\delta^2 \frac{1}{2-\delta} a^{x-2} dx$$

$$= 1 + 2\frac{a^2 - a^\delta}{(a-1)(2-\delta)\ln a}.$$

$\square$

As in the case of bidding, $R_{\delta,a}$ interpolates between two extreme strategies. Specifically, for $\delta = 0$, it has consistency and robustness that are both equal to $1 + (1+a)/\ln a$, which has a unique minimizer $q^*$, as in (30). In this case, the strategy is equivalent to the competitively optimal one.

In the other extreme, for $\delta \to 2$, simple limit evaluations show that $\text{rob}(R_{2,a}) = 1 + 2a^2/(a-1)$, whereas $\text{cons}(R_{2,a}) = 1 + 2/(a-1)$. From the study of deterministic search strategies [5], this is the best possible consistency/robustness tradeoff that can be attained by a deterministic strategy.

**Lower bound**   We show how to extend the approach of Section 3.2 to the linear search problem. Recall that, without predictions, the optimal competitive ratio is expressed by (30). For the proof that $q^*$ is optimal, we refer to Theorem 8.6 in [3], which is based on a distribution on targets with a hyperbolic pdf very similar to the online bidding case: the main difference is that targets may hide in either half-line, so the pdf is divided by a factor of 2, to reflect this. Using the same distribution on targets, and a prediction for a target at distance $\hat{u} = R$, we have the following expressions on the robustness and the consistency of a deterministic strategy $X = ((-1)^i x_i)_{i=0}^n$.

$$\text{rob}(X) = 1 + \varepsilon n + \sum_{i=0}^{n-1} \frac{x_i}{x_{i-1}}$$

$$\geq 1 + \varepsilon n + \varepsilon n \prod_{i=0}^{n-1} \frac{x_i}{x_{i-1}}$$

$$= 1 + \varepsilon n + \varepsilon n R^{1/n}$$

$$= 1 + \varepsilon n + \varepsilon \frac{R^{1/n}}{\ln R^{1/n}}$$

$$= 1 + (1-\varepsilon)\frac{1 + R^{1/n}}{\ln R^{1/n}}. \tag{31}$$

and

$$\text{cons}(X) = \frac{1}{R}(R + 2x_{n-2}) = 1 + 2\frac{x_{n-2}}{x_{n-1}}. \tag{32}$$

The above expressions are the counterpart of (15) and (14), respectively, for the linear search. Eq (31) follows directly from the proof of [3], whereas (32) follows by assuming, without loss of generality, that the searcher locates the target at distance $R$ at the $n-1$-th iteration. This is because the target will be located either at the $n$-th or the $(n-1)$-th iteration, and the latter minimizes the cost, hence the consistency as well.

With this setup in place, we obtain the following lemma, whose proof follows the same lines as Lemma 12.

**Lemma 16.** *Given $\varepsilon > 0$, let $T$ denote the quantity $\frac{R^{\frac{1}{n-1}}}{\ln R^{\frac{1}{n-1}}}$. For any $\lambda > 0$, it holds that*

$$\text{rob}(X) + \lambda \cdot \text{cons}(X) \geq 1 + \varepsilon n + (1-\varepsilon)T + \lambda(1 + \frac{1}{TW_0(T)}) - \delta(\varepsilon)$$

*where $W_0$ is the principal branch of the Lambert function, and $\lim_{\varepsilon \to 0} \delta(\varepsilon) = 0$.*

Last, Lemma 16 allows us to show the following lower bound.

**Theorem 17.** *Any randomized learning-augmented algorithm of robustness at most $r$ has consistency at least $1 + 2\frac{1}{(1+\xi)yW_0(y)}$, where $y = e^{W_0(\frac{r-1}{e})}$, for any arbitrarily small, but constant $\xi > 0$.*

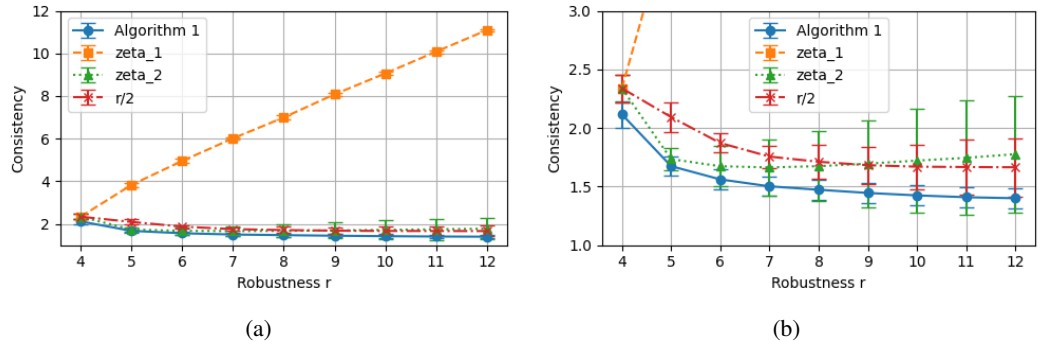

(a)                            (b)

Figure 4: Empirical consistency as a function of the robustness requirement $r$, where $\mu_i$ and $p_i$ are drawn i.i.d. following uniform distributions. Fig.(a) depicts the same values as (b), but has a wider $y$-axis so as to illustrate the relative performance of the heuristic with base $\zeta_1$.

*Proof sketch.* The proof is analogously to that of Theorem 12, with the difference that $S$ and $T$ are defined slightly differently, namely

$$S = \frac{R^{\frac{1}{n}}}{\ln R^{\frac{1}{n}}} \quad \text{and} \quad T = \frac{R^{\frac{1}{n-1}}}{\ln R^{\frac{1}{n}}}$$

Furthermore, from (31), the cases are defined relative to whether

$$1 + \frac{1 + R^{\frac{1}{n}}}{\ln R^{\frac{1}{n}}} \leq r,$$

i.e. based on whether $S \leq (1+\xi)e^{W_0(\frac{r-1}{e})}$. $\qquad\qquad\qquad\qquad\qquad\qquad\qquad\qquad\qquad\qquad\square$

## D   Further experimental results and discussion

In this section, we report additional experimental results.

Figure 4 depicts the empirical consistency of the algorithms for the dataset where $\mu_i$ and $p_i$ are drawn from uniform distributions defined in Section 5. Figure 4(a) highlights the empirical consistency of the heuristic $\zeta_1$. The plot shows that the consistency increases near-linearly with $r$, i.e., this heuristic is very inefficient. Hence we focus on the comparison to other algorithms. The results depicted in Figure 4(b) are comparable to the ones in the main paper for a spread Gaussian distribution. The standard deviation of the heuristic results is larger, as seen in the error bars, which is attributed to the fact that the uniform distribution has larger variance than the gaussian.

Figure 5 depicts experimental results for the remaining three datasets, in which all points have the same probability mass. Compared to datasets where $p_i$ are chosen i.i.d, the performance gap between the heuristics and Algorithm 1 is slightly smaller. The standard deviation of the heuristics is likewise smaller, since the i.i.d. predictions involve more randomness than the equiprobable predictions.

We conclude with a result that supports the experimental findings reported at the end of Section 4. Namely, we will show that for a large class of geometric heuristics, which includes those with bases $\zeta_2$ and $r/2$, the gap between their consistency and that of our algorithm becomes arbitrarily large, as the robustness $r$ increases.

Formally, we say that a geometric algorithm is *r-increasing* if it is of the form $\{\lambda \cdot \rho^i\}_{i \geq 0}$ where $\rho$ depends only on $r$, tends to $\infty$ as $r \to \infty$, and $\lambda$ may depend on $r$, $\rho$ and $\mu$. By definition, this class contains the heuristics with bases such as $r/2$, and $\zeta_2$.

Proposition 18 shows that $r$-increasing geometric algorithms may have an arbitrarily large consistency, even on a very simple prediction $\mu$ composed of two points. This is in contrast to Algorithm 1, whose consistency tends to 1, as $r \to \infty$.

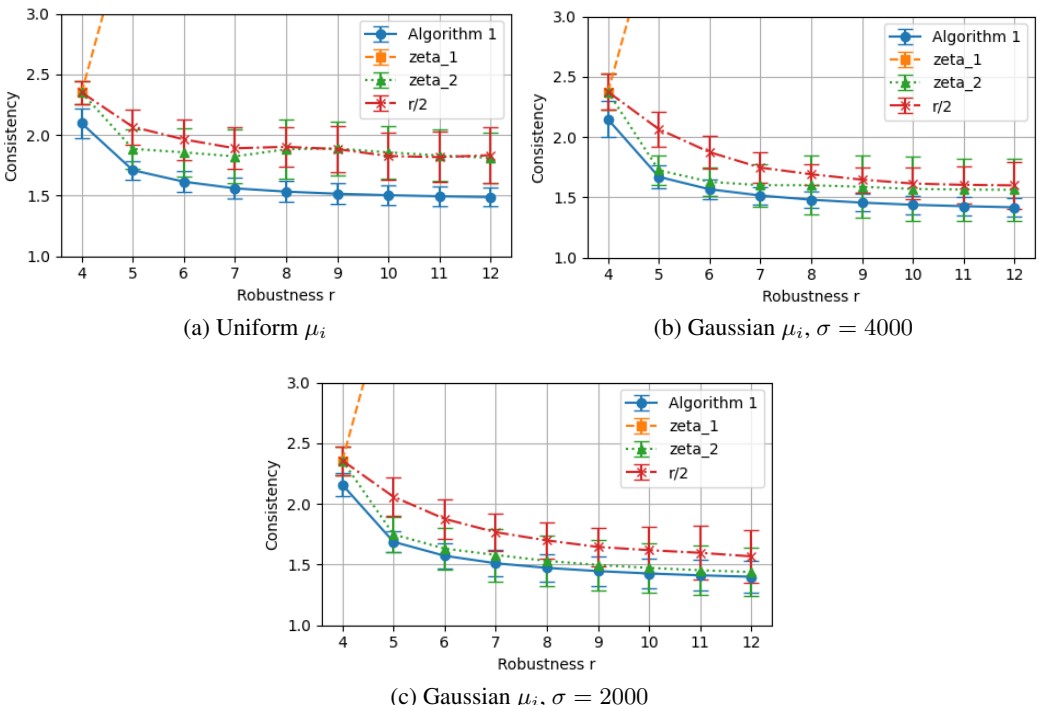

(a) Uniform $\mu_i$  (b) Gaussian $\mu_i$, $\sigma = 4000$

(c) Gaussian $\mu_i$, $\sigma = 2000$

Figure 5: Empirical consistency as a function of the robustness requirement $r$, when $p_i$ are equal, and the $\mu_i$ are drawn from the stated distributions.

**Proposition 18.** *For any $r$-increasing geometric algorithm, and any $R \geq 1$, there exists a robustness requirement $r$ and a prediction distribution $\mu$ over two points such that the consistency of the algorithm over $\mu$ is at least $R$, whereas the consistency of Algorithm 1 is at most $1 + 1/R$.*

*Proof.* Consider an $r$-increasing geometric algorithm, with base $\rho$ (which is a function of $r$). Let $\Delta = 2R$ and $r$ be such that $\rho \geq \Delta^2$ and $r \geq \Delta^2$; these choices are possible, by the definition of $r$-increasing strategies. Let $\mu$ be defined by $(\mu_1, \mu_2) = (1, \Delta)$ and $(p_1, p_2) = (1 - 1/\Delta, 1/\Delta)$. Since $\rho \geq \Delta^2$, there are only two possibilities for the sequence of bids $(x_i)_{i \geq 0}$ output by the algorithm: either there is a bid $x_i$ in $[\mu_1, \mu_2)$ or not. In the first case, $x_{i+1}$ is at least $\rho\mu_1 \geq \Delta^2$, and the expected cost of the algorithm is at least $p_2 \cdot \Delta^2 = \Delta$, by considering only the contribution of $\mu_2$ in the expected cost. In the second case, the first bid $x_j$ that is larger than $\mu_1$ is at least $\mu_2$, hence the expected cost of the algorithm over $\mu$ is at least $\mu_2 = \Delta$. In both cases, the expected cost of the algorithm is at least $\Delta = 2R$.

Consider now a partial strategy whose first two bids are equal to $\mu_1$ and $\mu_2$, respectively. This partial strategy is $r$-extendable, as $r \geq \Delta^2$ and the conditions of Lemma 6 are satisfied. Thus, there exists an $r$-robust strategy of expected cost equal to $(1 - \frac{1}{\Delta})\mu_1 + \frac{1}{\Delta}(\mu_1 + \mu_2) = 1 - \frac{1}{\Delta} + \frac{1}{\Delta} + 1 = 2$. Since $\mathbb{E}_{z \sim \mu}[z] = 2 - \frac{1}{\Delta}$, the above strategy has consistency at most $1 + 1/R$, and so does Algorithm 1 since it is Pareto-optimal. In contrast, the consistency of $X$ is at least $\frac{2R}{2 - 1/\Delta} \geq R$.   $\square$

The above results shows that any geometric strategy will have inferior performance: Either it is $r$-increasing, thus inefficient according to Proposition 18 or it is not, in which case it is inefficient even on single-point predictions. The latter follows from [8], which showed that for single-point predictions the consistency is equal to $\rho/(\rho - 1)$.

