# OpenReview forum: "Learning-Augmented Online Bidding in Stochastic Settings"
_NeurIPS.cc/2025/Conference — NeurIPS 2025 poster_

### Official Review · Reviewer_dCyV · 2025-06-08

**Clarity:** 3
**Significance:** 3
**Originality:** 3
**Rating:** 4
**Confidence:** 1

**Summary:**

This paper investigates the online bidding problem under a stochastic prediction oracle or stochastic algorithms. Given a stochastic prediction oracle, this paper proposes an algorithm to find the Pareto-optimal strategy. Moreover, this paper investigates the lower/upper bounds for stochastic algorithms.

**Questions:**

What is the potential difficulty of studying the stochastic algorithm with a stochastic prediction oracle?

**Ethical Concerns:**

["NO or VERY MINOR ethics concerns only"]

**Final Justification:**

From the latest discussion with the authors, the time complexity of Theorem 8 seems to be exponential with respect to the consistency approximation. However, I am not sure whether this is a big issue in this area. Therefore, I keep my rating but change the confidence to 1.

**Limitations:**

Yes

**Quality:**

3

**Strengths And Weaknesses:**

## Strengths
- The writing is clear.
- This paper gives a systematic study of online bidding in stochastic settings, including both upper bounds and lower bounds.
- Empirical results justify the efficiency of Algorithm 1.
- The paper gives an algorithm to find the entire Pareto frontier.

## Weaknesses
- There is no empirical comparison between the stochastic algorithm proposed in this paper and the deterministic algorithm.
- The computational complexity of Algorithm 1 is exponential in $k$.
- The experiments are restricted to some synthetic data, which cannot justify why the stochastic setting is reasonable.

---

> ### Author Rebuttal · Authors · 2025-07-27
>
> We are thankful to the reviewer for their constructive comments and helpful feedback.
>
> **Response to questions**
>
> 1. From the upper bound point of view, the difficulty is finding the structure of optimal solutions, a task that is already difficult even with one-side stochasticity, as shown in our LP-based approach. From the lower bound point of view, the difficulty is that the proof must be based on an entire family of adversarial distributions that should be tailored to the stochastic oracle; again this task is highly complex even for one adversarial distribution, as shown in the proof of our single-prediction bound.  One should also take into consideration that the competitive ratio of this problem is a constant, hence the margins for theoretical improvements are quite small. It should be emphasized that our lower bound of Section 3 shows that any improvement due to randomization quickly dissipates as $r$ increases. This implies that studying the full model will be a hard theoretical problem, but will not lead to significant improvements to our Algorithm 1, under a worst-case analysis. To make such a future study more accessible, one could start with a limited randomness model, in which the algorithm has access to only a few random bits: this limited randomness can be incorporated into the LP, at the expense of a much more complex proof approach.
>
> **Additional comments**
>
> 1. Our main contribution in Section 3 is the lower bound of Theorem 13, which shows that the improvement that can be obtained from randomization becomes marginal, as $r$ increases.
>
> 2. In regards to the complexity of Algorithm 1: If $k$ is too large, then Theorem 8 shows how to reduce complexity via distributional quantization, thus obtaining improved, non-trivial runtime guarantees. We also refer to the discussion in lines 181-183. The runtime analysis in this theorem depends on the dynamic range of the distribution, but is independent of $k$.
>
> 3. There are no established real benchmarks for online bidding, hence we used synthetic data which suffice to show a large performance gap between our algorithm and state-of-the-art heuristics, considering that our Algorithm 1 is theoretically optimal on *any* distribution. We note that even for concrete applications that can be modeled as the online bidding problem, the experimental analysis is often based on robust synthetic data. Consider, e.g., the problem of job submissions to busy data centers, such as the Amazon AWS computing platform, which can be modeled as an application of online bidding (See point 4 in the response to Reviewer C7xM for a formal definition of the setting). In [Aupy et al, 2019], the experimental analysis is based, likewise, on synthetic data, using distributions that can adequately simulate real settings. Our aim was to obtain an algorithm of optimal performance under any current, or future stochastic oracle that may become available. This is the broader objective in the emerging subfield of learning-augmented algorithms with distributional predictions.
>
> [Aupy et al, 2019] Aupy, Guillaume, Ana Gainaru, Valentin Honoré, Padma Raghavan, Yves Robert, and Hongyang Sun. "Reservation strategies for stochastic jobs." In 2019 IEEE International Parallel and Distributed Processing Symposium (IPDPS), pp. 166-175. IEEE, 2019.

---

> ### Comment · Reviewer_dCyV · 2025-08-01
>
> Thank you for your response! I have another question regarding the time complexity of Theorem 8.
>
> - The consistency approximation is $e^{1/c}$.
> - From the proof of Theorem 8, $\prod_{i=2}^k (j_i-j_{i-1})\leq (\frac{M}{m})^{2c\log(4C/c)}$, which implies that the time complexity is polynomial to $(\frac{M}{m})^{2c\log(4C/c)}$.
> - By letting $\epsilon = e^{1/c} - 1>0$, we can see that
>   - The consistency approximation is $1+\epsilon$.
>   - The time complexity is polynomial to $(\frac{M}{m})^{\frac{2}{\log(1+\epsilon)}\log(4C/c)}\geq (\frac{M}{m})^{\frac{2}{\epsilon}\log(4C/c)}$, which is exponential in $\frac{1}{\epsilon}$.

---

> > ### Author Response · Authors · 2025-08-02
> >
> > Thank you very much for your response and the feedback. We assume that you are asking for a comment on your observation. Indeed, the complexity is exponential in $1/\epsilon$, but this is consistent with both the statement of Theorem 8 and our discussion in lines 182-183: Namely, as long as the dynamic range is reasonably small (polynomial in the size of the instance) we obtain a polynomial time approximation scheme (PTAS). That is, for any *fixed*, constant $\epsilon>0$, the running time is polynomial and the consistency is equal to $1+\epsilon$. This is already an enormous improvement relative to the known heuristics, which have an unbounded consistency as a function of $r$ even on extremely simple distributions, as shown in Proposition 18.

---

> ### Comment · Reviewer_dCyV · 2025-08-02
>
> I will discuss with the AC and other reviewers about that. Thank you for your response!

---

> > ### Author Response · Authors · 2025-08-06
> >
> > Thank you very much for your response, and for the positive evaluation on all individual metrics. If there are any further details we could clarify, in the meantime, we would be happy to do so.

---

### Official Review · Reviewer_C7xM · 2025-06-19

**Clarity:** 2
**Significance:** 2
**Originality:** 3
**Rating:** 3
**Confidence:** 1

**Summary:**

In this paper, the authors studied the online bidding problem in an learning-augmented setting. The paper comprises of two parts. In the first part, the authors first analyzed online bidding with distributional predictions, identifying Pareto-optimal algorithms that optimally balance consistency and robustness. In the second part, they then investigated randomized bidding algorithms, providing upper and lower bounds on their performance tradeoffs.

**Questions:**

See weaknesses.

**Ethical Concerns:**

["NO or VERY MINOR ethics concerns only"]

**Final Justification:**

I appreciate the authors' response. I will keep my score due to concerns related to the practicality of the algorithm and I feel that the paper lacks runtime results from empirical evaluations. Nonetheless, I'm also open to discussions with the AC and other reviewers.

**Limitations:**

See weaknesses.

**Quality:**

3

**Strengths And Weaknesses:**

Strength:
- The paper builds on prior work in online bidding, which primarily focused on deterministic settings. It offers a comprehensive discussion by incorporating stochasticity in both the prediction oracle and the algorithm. The theoretical results appear sound, although I have not checked the proofs in detail.

Weakness:
- For Section 2, it would be helpful if the authors could elaborate on the main challenges or novel aspects of their proof techniques. As presented, the theoretical results are somewhat hard to digest.
- The paper notes that a practical implementation of Algorithm 1 may become computationally expensive as $k$ grows, and proposes a quantization rule to address this. Have the authors implemented this approach in practice? If so, a runtime comparison would be informative.
- From the numerical experiments, it seems that both the $\zeta_2$ and $r/2$ heuristics perform reasonably well in terms of consistency, albeit slightly worse than the proposed algorithm. Given that these heuristics are likely much easier to implement and computationally cheaper, it would be helpful if the authors could provide some discussion comparing their practical tradeoffs.
- Overall, while the theoretical results are interesting, the paper currently lacks a clear discussion of how the insights from these results translate to practical relevance. It would be helpful if the authors could comment on how the theoretical intuitions might inform or impact real-world applications.

---

> ### Author Rebuttal · Authors · 2025-07-27
>
> We are thankful to the reviewer for their constructive comments and helpful feedback.
>
> **Response to questions**
>
> 1. The main challenge in Section 2 is that, unlike the deterministic setting where simple geometric strategies of the form $(\lambda b^i)_{i \geq 0}$ suffice, in the stochastic setting such strategies can be extremely inefficient even on trivial distributions. Hence one needs to search a much larger space of candidate strategies which have no obvious structure. We introduce a novel approach based on Linear Programming  formulations to achieve this goal. But to do so, we need to address two technical difficulties: 1) Prune the solution space: this is achieved by using "tight" strategies as defined in (5), and by relying on Proposition 2; 2) Show necessary and sufficient conditions so as to extend a partial, finite strategy to a full, infinite strategy with robustness guarantees : this is accomplished in Lemmas 5 and 6. All these are novel approaches that have applications to other settings, such as contract scheduling with dynamic predictions and search games for a hidden target, as we show in Section 4.
>
> 2. Theorem 8 provides strict, theoretical guarantees both on the performance of the quantized algorithm and its runtime, where the latter does not depend anymore on $k$. As we note in lines 181-183, in practical settings when we expect the distributional advice to have reasonably small dynamic range (otherwise the oracle gives very inefficient information), the algorithm has provably fast runtime guarantees.
>
> 3.  Figure 3 shows that even for very simple distributions over only two points, the  $\zeta_2$ and $r/2$ heuristics can be as bad as almost 2-consistent, whereas our solution is 1-consistent, i.e., they are almost 100% worse than our solution. But the improvement we obtain can be much more dramatic, as we mention in lines 337-338, and as we show in Proposition 18 in the Appendix: namely as $r$ increases, the gap between these heuristics and our optimal solution is provably *unbounded*, even on distributions defined over two points. Hence, the two heuristics are provably *very inefficient* even for simple, yet adversarial data sets.
>
> 4.  For a concrete application of online bidding, consider the example of job submissions to busy data centers, such as the Amazon AWS computing platform. A user submits a job to be processed by the platform, but does not know in advance how much time it will require for its completion. The user thus proposes a *walltime* $t$, and pays cost equal to $t$. If the job is not completed within time $t$, the user must resubmit the job with a new walltime $t'>t$, and the total cost payed by the user is the sum of all walltimes until the job is finally completed. How should the user decide the sequence of submitted walltimes so as to minimize the ratio between the cost it pays, and the optimal cost that would be paid if the required time was known to the user in advance? This is precisely the formulation of online bidding, and it is at the core of scheduling problems in cloud computing, see for example [Aupy et al, 2019] .
>
> We will add a discussion of this application, as an example of the many applications we already mentioned in lines 32-37 and  Section 4 (that is, interruptible systems, search problems,  incremental clustering etc).
>
>
> [Aupy et al, 2019] Aupy, Guillaume, Ana Gainaru, Valentin Honoré, Padma Raghavan, Yves Robert, and Hongyang Sun. "Reservation strategies for stochastic jobs." In 2019 IEEE International Parallel and Distributed Processing Symposium (IPDPS), pp. 166-175. IEEE, 2019.

---

> > ### Comment · Reviewer_C7xM · 2025-08-03
> > **Thank you for your response**
> >
> > Thank you for your detailed response! Following up on my #2 above, I'm wondering if the authors can provide more details related to the quantization rule that results in Theorem 8. Has this rule been implemented and if so could you provide some related runtime results? I'm hoping to better understand the practicality of the proposed method.

---

> > > ### Author Response · Authors · 2025-08-03
> > >
> > > Thank you for your question. The quantization on which Theorem 8 is based follows an *exponential* rule, in that the quantized levels increase exponentially. This is explained formally in lines 860-864 in the proof (found in the appendix). This rule helps us remove the dependency on $k$, and obtain a time-efficient implementation as a function of the dynamic range of the distribution. Moreover, this quantization can be done in linear time in $k$ given the CDF of the distribution, so it is extremely fast. For this reason, the implementation focused on the $k$- point discrete setting (Algorithm 1).
> > >
> > > As a result, we obtain strict, worst-case, *theoretical* guarantees on the resulting performance of the algorithm, both in terms of the achieved consistency and the runtime. That is, as long as the dynamic range is polynomial in the size of the instance, then for any *fixed*, constant epsilon, the running time is polynomial and the consistency is equal to $1+\epsilon$. The practical implication is that we obtain an enormous improvement relative to the known heuristics, which have an unbounded consistency as a function of $r$ even on extremely simple distributions, as shown in Proposition 18.

---

### Official Review · Reviewer_88Mf · 2025-06-29

**Clarity:** 3
**Significance:** 3
**Originality:** 3
**Rating:** 5
**Confidence:** 3

**Summary:**

The paper considers the online bidding problem, in which an agent submits a sequence of bids until the bid exceeds an unknown target threshold. The goal is to minimize the sum of bids until the target threshold is reached. For this problem, the best–possible competitive ratios are $4$ for deterministic algorithms and $e$ for randomized algorithms.

In this paper, the authors consider two learning-augmented settings for online bidding. First, they consider deterministic algorithms with access to distributional predictions, i.e., predictions on the distribution from which the unknown target threshold is drawn. Then, they consider randomized algorithms with access to non-distributional predictions, i.e., a single value prediction on the unknown target threshold.

For the first setting, the authors give algorithms that achieve the best-possible consistency for each robustness requirement. Here, the consistency of an algorithm is the competitive ratio (of expectations) that the algorithm achieves if the threshold is indeed drawn from the predicted distribution and the robustness is the competitive ratio for arbitrary thresholds. The algorithm is based on first computing a partial strategy (finite bidding sequence) that is consistent, and robust if the finite bidding sequence reaches the unknown threshold. Then, they show how to extend the partial strategy to a full one that achieves the target robustness even if the partial sequence does not reach the unknown thresholds. The main technical contribution here is a characterization of robust and extendentable partial strategies. With the help of this characterization, the pareto-optimal strategy can be computed by solving a family of linear programs.

In the second setting, the authors give lower and upper bounds on the best-possible trade-off between consistency and robustness. Interestingly, the results show that the benefit of randomization is largest for small robustness requirements and decreases with increasing requirements.

**Questions:**

* Do you have any insights on the brittleness of your algorithm? I.e., what happens if the predicted distribution is only slightly wrong?
* Can you elaborate on the choice of $k=4$ in the empirical section? What is the reasoning behind fixing $k$ at $4$?
* Can you elaborate on the main new technical ideas in your paper compared to previous works on (learning-augmented) online bidding?

**Ethical Concerns:**

["NO or VERY MINOR ethics concerns only"]

**Final Justification:**

In my opinion, the paper is a strong contribution. The study of distributional predictions is a trending topic in learning-augmented algorithms, making this paper highly relevant. As outlined in my original review, the paper characterises the complete Pareto-frontier for the trade-off between consistency and robustness for deterministic algorithms with distributional predictions, and gives a very nice lower bound for randomised algorithms with deterministic predictions. My main concern regarding the brittleness has been adequately addressed in the author's rebuttal. Hence, I would like to increase my estimation from 4 to 5.

I do not share the concerns about the running time of the presented algorithms. While I don't have any insights on whether the running time is practical or not, I think that PTAS-type guarantees as given in Theorem 8 are well established from a theoretical point of view. In my opinion, resolving the Pareto-frontier is a strong contribution either way.

**Limitations:**

yes

**Quality:**

3

**Strengths And Weaknesses:**

Strength:
* The considered prediction model of distributional predictions seems highly relevant and has, thus far, received significantly less attention than non-distributional predictions. More positive results on these types of predictions could be impactful for the future direction of the field.
* The presented results on distributional predictions resolve the question for deterministic pareto-optimal algorithms along the complete pareto frontier.
* Even though it might not be the main result of the paper, I found the insights on randomized algorithms interesting. In particular, the observation that the benefit of randomization decreases with an increasing robustness requirement is very nice. To my knowledge, results of this type are not very common in the learning-augmented algorithm field.

Weaknesses:
* The paper is missing a discussion of error measures: What happens if the unknown threshold is not drawn from $\mu$ but from a distribution “similar” to $\mu$? Does the algorithm immediately revert to the robustness case or are improved guarantees still possible?
* The choice of the parameter $k=4$ in the empirical part of the paper seems a bit arbitrary. It is not clear to me if this choice is a good representative of distributional predictions that might show up in practice. Similar to the previous point, experiments where the threshold is drawn from a slightly different distribution would be quite interesting.
* The algorithms of Theorem 7 and 8 have an exponential resp. pseudo polynomial running time. A formal justification of these running times, for example in the form of hardness results, is missing.
* Figure 1 in Section 3 compares the deterministic upper bound with the randomised lower bound. It would be nice to also integrate the upper bound of Theorem 10 to put the algorithmic results into perspective.

---

> ### Author Rebuttal · Authors · 2025-07-27
>
> We are thankful to the reviewer for their constructive comments and helpful feedback.
>
> **Response to questions**
>
> 1. Brittleness is mainly a byproduct of "single-point" prediction oracles, combined with deterministic algorithms and deterministic analysis. In stochastic settings, brittleness intuitively is much less of an issue, as was shown formally in [20]. Notwithstanding this observation, there are various ways we can enhance Algorithm 1 to make it more robust to prediction errors. First, note that we can detect a potentially brittle solution by observing whether any of the bids is very close to one of the $k$ probability masses. Then we can rerun the algorithm using one of the following two approaches: either based on a "tolerance" parameter $\delta$, in which each of the $k$ "critical" bids must be at least a function of $\delta$ larger than the corresponding $\mu_i$ (this action can be encoded in the LP); or by introducing some randomized "perturbation" of the solution, and specifically of the size of the $k$ critical bids, along the lines of [20]. In what concerns the randomized setting, the strategy of Theorem 10 is inherently more robust to brittleness thanks to randomization.
>
>     Furthermore, in regards to error measures, one may be able to obtain an EMD-based analysis, by generalizing the approach of [7], which studied the contract scheduling problem with distributional predictions for robustness $r=4$. But to obtain a general result for Pareto-optimal algorithms will require some realistic assumptions on the distribution (e.g., that is has continuous, smooth and bounded density). Otherwise, we know from [7] that any Pareto-optimal algorithm is brittle against single-point predictions, and also against  single-point distributional predictions with very small error (measured by the EMD).
>
> &nbsp;
>
> 2. Considering that our algorithm is guaranteed to be optimal on *all* distributions, our goal was to evaluate it, in comparison to known baselines, on distributions that do not need to be overly complex. As shown in the experiments, even a relatively small value such as $k = 4$ allows us to draw very useful conclusions about the comparative performance of all algorithms.
>
> &nbsp;
>
> 3. Compared to previous work that focused on deterministic models and algorithms, our main novel technical ideas are as follows:
>
> - For distributional predictions (Section 2): The main challenge in Section 2 is that, unlike the deterministic setting where simple geometric strategies of the form $(\lambda b^i)_i$ suffice, in the stochastic setting such strategies can be extremely inefficient even on trivial distributions. Hence one needs to search a much larger space of candidate strategies which have no obvious structure. We introduce a novel approach based on Linear Programming formulations to achieve this goal. But to do so, we need to address two other technical points: 1) Prune the solution space: this is achieved by using "tight" strategies as defined in (5), and by relying on Proposition 2; 2) Show necessary and sufficient conditions so as to extend a partial, finite strategy to a full, infinite strategy with robustness guarantees : this is accomplished in Lemmas 5 and 6. All these are novel approaches that have applications in other settings such as contract scheduling with dynamic predictions and search games for a hidden target, as we show in Section 4.
>
>
> - For randomized algorithms (Section 3), the main challenge is the lower bound for two reasons: First, Yao’s principle (the main tool for randomized lower bounds) does not immediately apply, since we have two objectives in a tradeoff relation; Second, while deterministic $r$-robust strategies have a very clear characterization and known properties from previous work, there is no known characterization of the structure of  *randomized* $r$-robust strategies. We introduce two novel ideas:  A lower bound on a linear combination of the objectives ($rob+\lambda \cdot cons$), in Theorem 12, which allows us to give lower bounds on the trade-off between the two objectives; and an explicit, adversarial target distribution that allows us to properly quantify both consistency and robustness. In contrast, past work on randomized bidding without prediction used a very complex approach based on LP duality [26] which cannot incorporate predictions in any obvious way. Last, concerning the upper bound of Theorem 10, the novelty is to trade randomness for variability in performance: this allows us to obtain actual consistency/robustness tradeoffs, whereas with full randomness an algorithm has the same performance across all targets, and cannot use the prediction to the benefit of the consistency.

---

> ### Comment · Reviewer_88Mf · 2025-08-04
>
> Thank you to the authors for their reply, and in particular for their helpful and interesting comments on the brittleness of the algorithm.

---

### Official Review · Reviewer_M7pr · 2025-07-02

**Clarity:** 4
**Significance:** 3
**Originality:** 3
**Rating:** 5
**Confidence:** 4

**Summary:**

The authors study the online bidding problem where we submit a sequence of bids until one bid exceeds a fixed but hidden value. The goal is to minimize the total sum of bids (including the final bid that exceeds the hidden value). The problem is typically studied under competitive analysis, and the authors transfer this to the learning-augmented setting where we want to bound the consistency (which describes how well we are doing compared to the prediction) and robustness (which is still the competitive ratio).

The paper contains two algorithms for different settings. In the first setting, a stochastic prediction with finite support is provided to the algorithm. Through a structural analysis of the problem, the authors manage to provide a Pareto-optimal algorithm which for any given robustness outputs a sequence of bids that achieve the best possible consistency.

The second algorithm is in the setting where the prediction is a fixed value, but the algorithm is allowed to run a randomized sequence and attain robustness and consistency in expectation. As in the prior work in the non-learning-augmented setting, this allows for slightly improved results. The authors extend these prior methods to the learning-augmented setting.

Finally, they complement their algorithms with an experimental evaluation.

**Questions:**

As it stands, the algorithm considers only two cases, namely when the prediction is totally accurate or totally adversarial. But it would be interesting to obtain guarantees that smoothly trade off the two cases, e.g., as a function of the EMD.

**Ethical Concerns:**

["NO or VERY MINOR ethics concerns only"]

**Final Justification:**

As outlined in my review, I believe the problem and solution is interesting and worth studying and the paper well written. Since the techniques are not groundbreaking, I am giving a score of 5.

**Limitations:**

yes

**Paper Formatting Concerns:**

-

**Quality:**

3

**Strengths And Weaknesses:**

### Strengths

1. The paper provides a complete contribution as they provide upper and lower bounds for the settings they consider. Especially in learning-augmented algorithms, such theoretical results are valuable as they are typically hard to obtain.
2. The authors provide fundamental theoretical contributions to the online bidding problem (such as Definition 1, Proposition 2, or Lemma 6) which may be valuable beyond learning-augmentation.
3. The paper is very well-written and easy to follow.

### Weaknesses

1. The algorithms do not address the setting where the prediction is stochastic and the algorithm is randomized, but this seems like a natural extension.
2. Experiments on real-world data would be great.

---

> ### Author Rebuttal · Authors · 2025-07-27
>
> We are thankful to the reviewer for their constructive comments and helpful feedback.
>
> **Response to questions**
>
> 1. It may indeed be possible to obtain such guarantees via an EMD-based analysis, by generalizing the approach of [7], which studied the contract scheduling problem with distributional predictions for robustness $r=4$. But to obtain a general, theoretical result for Pareto-optimal algorithms will require some realistic assumptions on the distribution (e.g., that is has continuous, smooth and bounded density). Otherwise, we know from [7] that any Pareto-optimal algorithm is brittle against single-point predictions, and also against single-point distributional predictions with very small error (measured by the EMD). While this is an interesting theoretical problem, we should note that the consistency/robustness tradeoffs that one obtains from distributional prediction analysis are much more robust, in practice, than single-point oracles and "deterministic" analysis.
>
> Concerning the future study of an extension with double-sided stochasticity, there are several challenges. From the upper bound point of view, the difficulty is finding the structure of optimal solutions, a task that is already difficult even with one-side stochasticity, as shown in our LP-based approach. From the lower bound point of view, the difficulty is that the proof must be based on an entire family of adversarial distributions that should be tailored to the stochastic oracle; again this task is highly complex even for one adversarial distribution, as shown in the proof of our single-prediction bound. But in the context of our lower bound of Theorem 13,  any improvement due to randomization quickly dissipates as $r$ increases. This implies that studying the full model will be a hard theoretical problem, but will not lead to significant improvements to our Algorithm 1, under a worst-case analysis. To make such a future study more accessible, one could start with a limited randomness model, in which the algorithm has access to only a few random bits: this limited randomness can be incorporated into the LP, at the expense of a much more complex proof approach.

---

> > ### Comment · Reviewer_M7pr · 2025-08-05
> >
> > Thank you for your response. Since I didn't have any major concerns, I will maintain my score.

---

> > > ### Author Response · Authors · 2025-08-06
> > >
> > > Thank you very much for your response, we appreciate that the rebuttal was helpful towards the evaluation. If they are any remaining concerns, even if minor, we would be very happy to address them.

---

### Decision · Program_Chairs · 2025-09-17

**Decision:**

Accept (poster)

**Comment:**

This paper studies an online bidding problem where the bidders submit a sequence of bids until one bid exceeds a fixed but hidden value. The goal is to propose a learning-augmented algorithm that not only achieves a high competitive ratio (robustness) but also a a good performance when the prediction is accurate (consistency).

Two settings are considered in this paper. In the first setting, a stochastic prediction with finite support is provided to the algorithm. A Pareto-optimal algorithm is proposed to output a sequence of bids that achieve the best possible consistency for any given robustness. In the second setting, the prediction is a fixed value, but the algorithm is allowed to be randomized. The authors extend prior methods in the non-learning-augmented setting to the learning-augmented setting and show improved results. Finally, the algorithms are complemented with an experimental evaluation.

All reviewers are unanimously supportive of this paper. It studies an interesting problem and provides sound solutions. Some concerns include the running time of the algorithm (e.g. exponential in 1/\varepsilon, possibly with a large degree in the polynomial-time), and that the contributions in the second setting is a bit incremental. Therefore, I would like to recommend acceptance as a poster.